# UME-R1: Exploring Reasoning-Driven Generative Multimodal Embeddings

**Zhibin Lan**[1]* **Liqiang Niu**[2] **Fandong Meng**[2] **Jie Zhou**[2] **Jinsong Su**[1,3,4]†

[1]School of Informatics, Xiamen University, China,
[2]WeChat AI, Tencent Inc, China,
[3]Key Laboratory of Digital Protection and Intelligent Processing of Intangible Cultural Heritage of Fujian and Taiwan (Xiamen University), Ministry of Culture and Tourism, China,
[4]Shanghai Artificial Intelligence Laboratory, China

`lanzhibin@stu.xmu.edu.cn`, `jssu@xmu.edu.cn`
`{poetniu, fandongmeng, withtomzhou}@tencent.com`

## ABSTRACT

The remarkable success of multimodal large language models (MLLMs) has driven advances in multimodal embeddings, yet existing models remain inherently discriminative, limiting their ability to benefit from reasoning-driven generation paradigm. In this work, we pioneer the exploration of reasoning-driven generative embeddings, unifying embedding tasks within a generative paradigm. We propose UME-R1, a universal multimodal embedding framework consisting of a two-stage training strategy: a cold-start supervised fine-tuning equips the model with reasoning capabilities and enables it to generate both discriminative and reasoning-driven generative embeddings; a subsequent reinforcement learning enhances reasoning and further optimizes generative embedding quality. This pioneering work reveals four key insights: 1) reasoning-driven generative embeddings unlock substantial performance gains over conventional discriminative embeddings by leveraging the powerful generative reasoning capabilities of MLLMs; 2) discriminative and reasoning-driven generative embeddings are complementary, whose combined oracle performance far exceeding that of either alone; 3) RL can effectively enhance reasoning-driven generative embeddings, establishing a scalable optimization paradigm; 4) repeated sampling at inference boosts downstream task coverage (pass@k), highlighting the inference-time scalability potential of reasoning-driven generative embeddings. Evaluated on the MMEB-V2 benchmark across 78 tasks spanning video, image, and visual documents, UME-R1 significantly outperforms conventional discriminative embedding models and offers a foundation for more interpretable, reasoning-driven generative multimodal embeddings. Our datasets, models, and code are available at `https://github.com/XMUDeepLIT/UME-R1`.

## 1 INTRODUCTION

Recently, the field of multimodal embeddings has been significantly advanced by the remarkable success of multimodal large language models (MLLMs). For instance, VLM2Vec (Jiang et al., 2025) and MM-Embed (Lin et al., 2025a) construct multimodal embedding models based on MLLMs. These models demonstrate superior performance across a range of multimodal embedding tasks compared to traditional dual-encoder vision–language models like CLIP (Radford et al., 2021).

In parallel, large reasoning models (LRMs) represented by GPT-4o (Hurst et al., 2024) and DeepSeek-R1 (Guo et al., 2025) have made breakthroughs in complex reasoning. A distinctive feature of these models is the incorporation of the chain of thought (CoT) (Wei et al., 2022a), which elicits step-by-step reasoning paths and typically produces more accurate and interpretable outputs. Building on this success, recent works (Shen et al., 2025b; Hong et al., 2025a) have extended these

---

* This work was done when Zhibin Lan was interning at WeChat AI, Tencent Inc, China.
† Corresponding author.

advances to MLLMs, substantially enhancing their performance on various multimodal tasks. However, multimodal embedding models have derived limited benefit from these advances. The key reason is that existing MLLM-based multimodal embedding models are discriminative: they directly encode the multimodal input and extract the last token's final hidden state as the embedding, without generating any new tokens. Naturally, this raises the question: *How to make a multimodal embedding model act as a generative one?*

Several prior studies (Ouali et al., 2025; Yu et al., 2025a) have incorporated a next-token prediction loss in training multimodal embedding models, demonstrating that it preserves generative capabilities while enhancing discriminative performance. Nevertheless, these approaches merely introduce additional data and losses during training. Ultimately, at inference, they remain discriminative, as their embeddings are obtained by directly encoding the input without generating any intermediate content; we refer to these as *discriminative embeddings*.

In this paper, we propose UME-R1, a **u**niversal **m**ultimodal **e**mbedding framework that enables multimodal embedding models to produce either discriminative or reasoning-driven generative embeddings on demand. First, we construct a cold-start supervised fine-tuning (SFT) dataset by augmenting the original query–target pairs used for embedding training with intermediate reasoning and summaries. During training, the contrastive loss is applied to embedding tokens that follow the summary, while an autoregressive next-token prediction loss is imposed on the reasoning and summary tokens. As a result, the model learns to first generate intermediate reasoning and a summary, and then produce embedding token to obtain representation; we term these as *reasoning-driven generative embeddings*. Meanwhile, discriminative embeddings are preserved throughout training, allowing the model to flexibly output either type of embedding as needed. Interestingly, experiments reveal a substantial gap between the oracle upper bound and current discriminative embeddings, indicating that there remains considerable room for improvement.

We further ask: *Can reinforcement learning with verifiable reward (RLVR) also be effective for generative embedding models?* A natural approach would assign a positive reward if the similarity of a given positive pair exceeds a preset threshold, and no reward otherwise. However, since the degree of similarity varies among different pairs, this approach may render some pairs excessively difficult or easy, resulting in the problem of zero policy gradients (Yu et al., 2025b). To overcome this, we propose a reward policy that considers ranking and similarity gaps simultaneously, and demonstrate that generative embedding models can also benefit from RLVR. Additionally, we find that repeated sampling can improve the coverage (i.e., pass@$k$) of generative embedding models, suggesting that embeddings also have the potential for inference-time scaling.

Overall, we make the following four contributions: ❶ Based on MMEB-V2 (Meng et al., 2025) training data, we build a multimodal embedding cold-start SFT dataset with CoT annotations, and construct a small-scale dataset for efficient RL training. ❷ We propose UME-R1, a framework designed to endow multimodal embedding models with the flexibility to switch between discriminative and reasoning-driven generative embeddings. To the best of our knowledge, we are the first to explore reasoning-driven generative embeddings, demonstrating the significant potential of unifying embeddings within a generative paradigm. ❸ We pioneer the successful application of rule-based RL to the multimodal embeddings task, which lacks standard best answers like math, by designing a novel reward policy tailored to embeddings. ❹ UME-R1 outperforms conventional discriminative embedding models on MMEB-V2, a benchmark comprising 78 tasks across three visual modalities: video, image, and visual documents. Analysis of an oracle upper bound and pass@$k$ indicates that UME-R1 retains significant potential for further improvement.

## 2 DATASET CONSTRUCTION

To construct the training corpus for generative multimodal embeddings, as illustrated in Figure 1, we sample 50,000 instances from each of the 20 in-distribution datasets within MMEB (Jiang et al., 2025). Following VLM2Vec-V2 (Meng et al., 2025), we also incorporate the training instances from LLaVA-Hound (Zhang et al., 2025a), ViDoRe (Faysse et al., 2025b), and VisRAG (Yu et al., 2025c) datasets to cover video and visual-document modalities, yielding a total of 1.76 million pairs. Subsequently, we employ the pure-thinking model GLM-4.1V-Thinking (Hong et al., 2025b) to generate CoT rationales for both the query and the target of each pair.

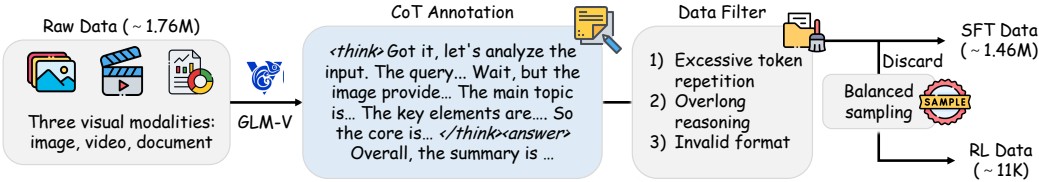

Figure 1: Illustration of the pipeline for data construction. Specific prompts used for CoT annotation and the resulting data samples are presented in Appendix C.

We filter the data by excluding pairs that meet any of the following criteria: (1) contain extensive contiguous token repetition; (2) include reasoning that are excessively long (e.g., exceeding 8,192 tokens); or (3) produce responses that do not conform to the `<think>...</think><answer>` format. This filtering process results in a final set of 1.46 million cold-start SFT pairs. For RL training, a set of 11,136 pairs is balanced sampled from various datasets spanning the image, video, and visual-document modalities, prioritizing instances not included in the SFT data to avoid overly simple samples.

## 3  UME-R1

### 3.1  PRELIMINARIES

We adopt the formulation from VLM2Vec (Jiang et al., 2025) for discriminative multimodal embeddings task as follows: given a query $q$ and its corresponding positive target $t^+$, as well as a set of negative targets $\mathcal{T}^- = \{t_1^-, \ldots, t_K^-\}$, the objective is to maximize similarity between $q$ and $t^+$ over all $q$ and $t^- \in \mathcal{T}^-$ pairs. Here, both queries and targets can be text, image, or interleaved text-image.

In practice, we sample a mini-batch of $N$ query–target pairs $(q_1, t_1), \ldots, (q_N, t_N)$, where $(q_i, t_i)$ forms the positive pair and all targets $\{t_j \mid j \neq i\}$ serve as negatives for $q_i$. Formally, we optimize the model by minimizing the following InfoNCE loss function:

$$\mathcal{L}_{dctr} = \frac{1}{N} \sum_{i=1}^{N} -\log \frac{\exp\big((\pi_\theta(q_i) \cdot \pi_\theta(t_i))/\tau\big)}{\exp\big((\pi_\theta(q_i) \cdot \pi_\theta(t_i))/\tau\big) + \sum_{j \neq i}^{N} \exp\big((\pi_\theta(q_i) \cdot \pi_\theta(t_j))/\tau\big)}. \quad (1)$$

where $\pi_\theta(\cdot)$ denotes the normalized representation of the last input token, derived from the MLLM's final-layer hidden state, and $\tau$ represents the temperature hyper-parameter.

### 3.2  ARCHITECTURE

In this work, we introduce a multimodal embedding model capable of producing both discriminative and reasoning-driven generative embeddings. To obtain the reasoning-driven generative embeddings, the model first generates distinct reasoning and summaries for each query and target. These outputs are then concatenated with the original input to produce the final generative representation. Note that the model can simultaneously yield discriminative embeddings without incurring additional computation. Specifically, we employ the following template to realize this process:

---

**Template for Discriminative and Reasoning-Driven Generative Embeddings**

*USER*: `<image> <video>` {`query/target`} `<disc_emb>`
*Represent the above input text, images, videos, or any combination of the three as embeddings. First output the thinking process in* `<think> </think>` *tags and then summarize the entire input in a word or sentence. Finally, use the* `<gen_emb>` *tag to represent the entire input.*
*ASSISTANT*: `<think>` {`reasoning`} `</think>`
`<answer>` {`summary`} `<gen_emb>`

---

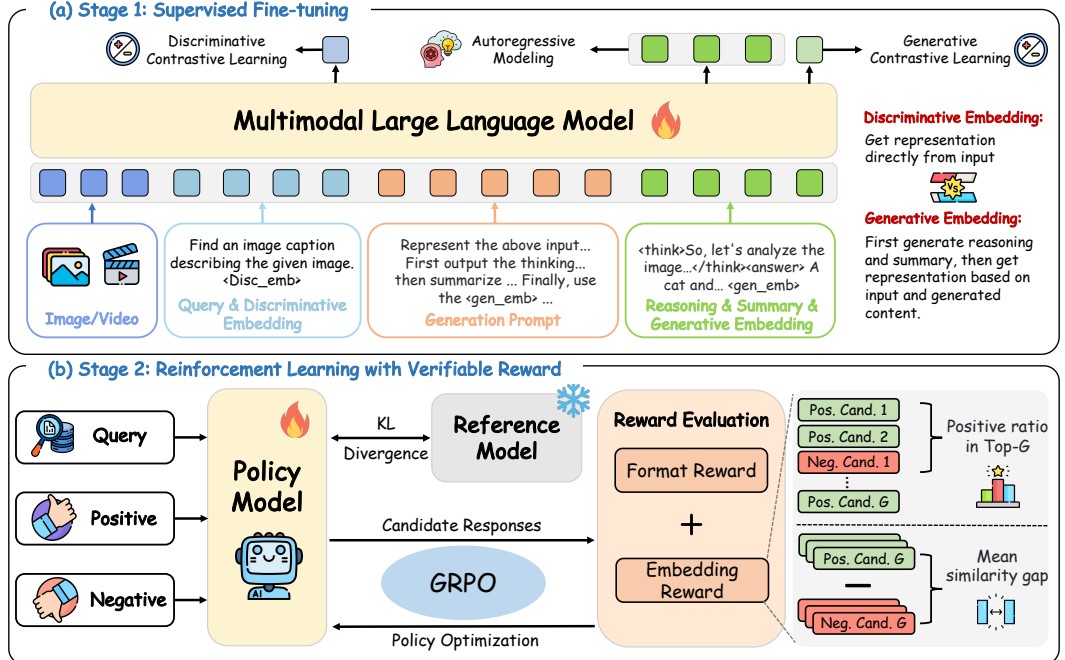

Figure 2: Overview of UME-R1. UME-R1 introduces a two-stage training framework for generative multimodal embedding. (a) Supervised fine-tuning uses query-target pairs with reasoning annotations to train the MLLM, enabling it to generate both discriminative and reasoning-driven generative embeddings as well as to possess basic reasoning abilities. (b) RLVR continues to fine-tune the model using regular query-target pairs, encouraging it to generate reasoning trajectories that lead to more beneficial embeddings.

where `<image>` and `<video>` denote placeholders for the input image and video. As illustrated in Figure 2(a), the last-layer hidden states corresponding to the prompt's `<disc_emb>` token and the final model-generated `<gen_emb>` token serve as the discriminative and reasoning-driven generative embeddings, respectively.

## 3.3 MODEL TRAINING

We train the model in two stages, enabling it not only to generate discriminative embeddings but also to develop reasoning capabilities for producing stronger reasoning-driven generative embeddings. Figure 2 illustrates the overall training process.

**Stage 1: Supervised Fine-tuning.** In this initial stage, we perform SFT on the model using the multimodal embedding dataset constructed in Section 2, which incorporates the step-by-step reasoning processes. As shown in Figure 2(a), alongside the discriminative embedding training objective outlined in Section 3.1, we also include the following generative embedding training objectives:

$$\mathcal{L}_{gctr} = \frac{1}{N} \sum_{i=1}^{N} -\log \frac{\exp\big((\pi_\theta(q_i, o_i^q) \cdot \pi_\theta(t_i, o_i^t))/\tau\big)}{\exp\big((\pi_\theta(q_i, o_i^q) \cdot \pi_\theta(t_i, o_i^t))/\tau\big) + \sum_{j \neq i}^{N} \exp\big((\pi_\theta(q_i, o_i^q) \cdot \pi_\theta(t_j, o_j^q))/\tau\big)}.$$
(2)

where $o_i^q$ and $o_i^t$ denote the $i$-th reasoning trajectory and summary of the query and target, respectively. Compared to the original input, reasoning process and summarization provide more detailed and useful information, which often enhances the performance of the resulting embeddings.

Furthermore, to endow the model with reasoning capabilities during inference, we apply a next-token prediction loss over both the reasoning trajectories and summaries, formalized as

$$\mathcal{L}_{ce} = -\frac{1}{N} \sum_{i=1}^{N} \left( \sum_{j=1}^{L_q} \log \pi_\theta \left( o_{i,j}^q \mid q_i, o_{i,<j}^q \right) + \sum_{j=1}^{L_t} \log \pi_\theta \left( o_{i,j}^t \mid t_i, o_{i,<j}^t \right) \right), \qquad (3)$$

where $L_q$ and $L_t$ denote the lengths of the reasoning trajectories for the query and the target, respectively. Overall, the loss for the SFT stage is defined as follows:

$$\mathcal{L}_{sft} = \mathcal{L}_{dctr} + \mathcal{L}_{gctr} + \mathcal{L}_{ce}. \qquad (4)$$

This stage of training not only equips the model to generate both discriminative and reasoning-driven generative embeddings, but also lays the foundation for its reasoning abilities.

**Stage 2: Reinforcement Learning with Verifiable Reward.** As illustrated in Figure 2(b), in this stage, we further refine the model $\pi_\theta$ using Group Relative Policy Optimization (GRPO) (Shao et al., 2024). Unlike methods that rely on a learned value function, GRPO utilizes the mean reward across multiple sampled outputs as its baseline. Specifically, for each input query $q$, it samples a group of $G$ candidate responses $\{o_i\}_{i=1}^{G}$ from the old policy $\pi_{\theta_{old}}$, and then optimizes the policy model $\pi_\theta$ by maximizing the following objective:

$$\mathcal{L}_{\text{grpo}} = \mathbb{E}_{q \sim \mathcal{D}, \{o_i\}_{i=1}^{G} \sim \pi_{\theta_{old}}} \left[ \frac{1}{G} \sum_{i=1}^{G} \left( \min \left( \frac{\pi_\theta(o_i \mid q)}{\pi_{\theta_{old}}(o_i \mid q)} A_i, \right. \right. \right.$$

$$\left. \left. \left. \text{clip} \left( \frac{\pi_\theta(o_i \mid q)}{\pi_{\theta_{old}}(o_i \mid q)}, 1 - \epsilon, 1 + \epsilon \right) A_i \right) - \beta \mathbb{D}_{KL}(\pi_\theta \| \pi_{\text{ref}}) \right) \right], \qquad (5)$$

where $\mathcal{D}$ denotes the training dataset, $\epsilon$ and $\beta$ are hyper-parameters, and $\pi_{ref}$ represents the reference model before optimization. $A_i$ indicates the advantage of the $i$-th response, computed based on a group of rewards $\{r_1, \ldots, r_G\}$ corresponding to the outputs within each group:

$$A_i = \frac{r_i - \text{mean}(\{r_1, \cdots, r_G\})}{\text{std}(\{r_1, \cdots, r_G\})}. \qquad (6)$$

Accordingly, we design the reward function to include two components: format rewards and embedding rewards, which we will now describe in detail.

*Format Reward.* The use of this reward encourages the model to adhere to a predefined template, ensuring that responses are well-structured and interpretable. Specifically, the model is required to perform reasoning within the `<think>` and `</think>` tags, provide a summary after the `<answer>` tag, and finally generate the `<gen_emb>` for obtaining the generative embedding. A reward of 1 is granted for strict adherence to the template, while any deviation results in a reward of 0.

*Embedding Reward.* This component is used to evaluate the quality of the embeddings generated by the model. Since embeddings cannot be directly evaluated against standard answers as in mathematics, we evaluate them from two aspects: the ranking of positives among negatives, and the similarity gap between positives and negatives. Concretely, for each query $q$ with a positive target $t^+$ and a negative target $t^-$, we sample a group of responses $\{o_j^+\}_{j=1}^{G}$ corresponding to the positive target, another group $\{o_j^-\}_{j=1}^{G}$ corresponding to the negative target[1]. For the $i$-th sampled response $o_i$ of the query, we calculate its similarity scores with the positive targets as $\mathcal{S}^+ = \{\pi_\theta(q, o_i) \cdot \pi_\theta(t^+, o_j^+)\}_{j=1}^{G}$, and with the negative targets as $\mathcal{S}^- = \{\pi_\theta(q, o_i) \cdot \pi_\theta(t^-, o_j^-)\}_{j=1}^{G}$. The embedding reward for the $i$-th response $o_i$ sampled from the query is defined as follows:

$$R_{emb}(o_i) = \underbrace{\frac{|\mathcal{S}^+ \cap \text{top}_G(\mathcal{S}^+ \cup \mathcal{S}^-)|}{G}}_{Ranking} \times \underbrace{\left( \text{avg}(\mathcal{S}^+) - \text{avg}(\mathcal{S}^-) \right)}_{Similarity\ Gap}, \qquad (7)$$

where $\text{top}_G(\cdot)$ denotes the operation of selecting the top-$G$ largest elements from input set. By optimizing this reward, the model learns to produce reasoning trajectories that are more conducive to generating high-quality generative embedding.

---

[1]For simplicity, only one negative target is illustrated; however, this method can extends to any number of negative targets in practice.

## 4 EXPERIMENTS

### 4.1 EXPERIMENTAL SETUP

**Training Details.** Following VLM2Vec-V2 (Meng et al., 2025), we adopt Qwen2-VL-2B and Qwen2-VL-7B as backbone models. During the SFT stage, we train using the cold-start dataset constructed in Section 2, which is approximately two-thirds the size of the dataset used by VLM2Vec-V2. Consistent with the settings of VLM2Vec-V2, the temperature $\tau$ is set to 0.02, the batch size to 1,024 (achieved through gradient accumulation), and the number of training steps to 5K. Besides, the maximum sequence length is 12,288 tokens, and the learning rate is 5e-5. During the RL stage, the model is trained on approximately 11K pairs and uses the default GRPO hyperparameter settings: group size $G = 8$, clipping parameter $\epsilon = 0.2$, and KL-divergence coefficient $\beta = 0.04$. In this stage, we set the batch size to 256, the learning rate to 1e-6, and train for one epoch.

**Evaluation.** We evaluate UME-R1 on MMEB-V2 (Meng et al., 2025), a benchmark that extends MMEB-V1 (Jiang et al., 2025) by introducing 5 meta-tasks focused on video and visual document, covering a total of 9 meta-tasks and 78 tasks. During inference, we use greedy search and set the maximum number of newly generated tokens to 8,192. Unless otherwise specified, we use reasoning-driven generative embeddings for evaluation. Hit@1 is used as the evaluation metric for all video and image tasks, while NDCG@5 (Järvelin & Kekäläinen, 2002) is reported for visual document tasks. In addition, we compare several strong models on MMEB-V1, with the corresponding results presented in Appendix E.

**Baselines.** We compare against several MLLM-based multimodal embedding models, including GME (Zhang et al., 2025b), ColPali (Faysse et al., 2025a), VLM2Vec (Jiang et al., 2025), LamRA (Liu et al., 2025a), CAFe (Yu et al., 2025a), and VLM2Vec-V2 (Meng et al., 2025). To ensure a fair comparison and to clearly assess the role of reasoning-driven generative embeddings, we evaluate a model that performs contrastive learning exclusively on discriminative embeddings, using the same dataset and settings as ours. We refer to this model as DUME (**d**iscriminative UME).

### 4.2 MAIN RESULTS

Table 1 presents a performance comparison between UME-R1 and the Baseline on 78 tasks spanning three visual modalities: images, videos, and visual documents. The results show that UME-R1 consistently achieves the best performance in images and videos with the same backbone. Although ColPali and GME perform well on visual document retrieval, the former is specifically optimized for visual document tasks, while the latter uses a large amount of closed-source data. In particular, compared to VLM2Vec-V2, UME-R1 achieves an overall improvement of 2.1 while using only two-thirds of its training data. Compared to the discriminative embedding model DUME trained with the same amount of data, UME-R1 increases the total scores for images, videos, and visual documents by 4.1, 9.0, and 11.1, respectively, fully demonstrating the effectiveness of reasoning-driven generative embeddings. Comparative examples of reasoning-driven generative and discriminative embeddings are provided in Appendix F.

Since UME-R1 can flexibly choose discriminative or reasoning-driven generative embeddings as needed, we report an oracle upper bound. For each test instance, the oracle selects the embedding mode that yields the best retrieval performance. Under the oracle setting, UME-R1-2B and UME-R1-7B achieve overall score improvements of 4.3 and 3.6, respectively. The results demonstrate that the oracle substantially outperforms using only reasoning-driven generative embeddings, which means that in practical applications users can freely switch modes to obtain more satisfactory retrieval results.

### 4.3 ABLATION STUDY

**Impact of RL Stage and Reward Design on Model Effectiveness.** As shown in Table 2, we study the effectiveness of different components in the RL stage across 78 tasks of MMEB-V2. From the second row, we observe that although the RL stage uses only a small dataset for training with GRPO and does not incorporate contrastive learning, it still substantially improves model performance. This finding suggests that effective reasoning paths and summarization contribute to better

Table 1: Comparison of performance between baselines and UME-R1 on MMEB-V2. **CLS**: classification, **QA**: question answering, **RET**: retrieval, **GD**: grounding, **MRET**: moment retrieval, **VDR**: ViDoRe, **VR**: VisRAG, **OOD**: out-of-domain. *Oracle* denotes the case where the best result between reasoning-driven generative and discriminative embeddings is picked. Detailed results can be found in Appendix D.

| Model | Image | | | | | Video | | | | | VisDoc | | | | | All |
|---|---|---|---|---|---|---|---|---|---|---|---|---|---|---|---|---|
| | CLS | QA | RET | GD | Overall | CLS | QA | RET | MRET | Overall | VDRv1 | VDRv2 | VR | OOD | Overall | |
| # of Datasets | 10 | 10 | 12 | 4 | 36 | 5 | 5 | 5 | 3 | 18 | 10 | 4 | 6 | 4 | 24 | 78 |
| *Baseline Models* | | | | | | | | | | | | | | | | |
| ColPali-V1.3 (PaliGemma-3B) | 40.3 | 11.5 | 48.1 | 40.3 | 34.9 | 26.7 | 37.8 | 21.6 | 25.5 | 28.2 | 83.6 | 52.0 | 81.1 | 43.1 | 71.0 | 44.4 |
| GME (Qwen2-VL-2B) | 54.4 | 29.9 | 66.9 | 55.5 | 51.9 | 34.9 | 42.0 | 25.6 | 32.4 | 33.9 | 86.1 | 54.0 | 82.5 | 43.1 | 72.7 | 54.1 |
| GME (Qwen2-VL-7B) | 57.7 | 34.7 | 71.2 | 59.3 | 56.0 | 37.4 | 50.4 | 28.4 | 38.2 | 38.6 | **89.4** | **55.6** | **85.0** | **44.4** | **75.2** | 57.8 |
| LamRA (Qwen2-VL-7B) | 59.2 | 26.5 | 70.0 | 62.7 | 54.1 | 39.3 | 42.6 | 24.3 | 34.6 | 35.2 | 22.0 | 11.5 | 37.4 | 21.0 | 23.9 | 40.4 |
| LamRA (Qwen2.5-VL-7B) | 51.7 | 34.1 | 66.9 | 56.7 | 52.4 | 32.9 | 42.6 | 23.2 | 37.6 | 33.7 | 56.3 | 33.3 | 58.2 | 40.1 | 50.2 | 47.4 |
| VLM2Vec (Qwen2-VL-2B) | 58.7 | 49.3 | 65.0 | 72.9 | 59.7 | 33.4 | 30.5 | 20.6 | 33.0 | 29.0 | 49.8 | 13.5 | 51.8 | 33.5 | 41.6 | 47.0 |
| VLM2Vec (Qwen2-VL-7B) | 62.7 | 56.9 | 69.4 | 82.2 | 65.5 | 39.1 | 30.0 | 29.0 | **40.6** | 34.0 | 56.9 | 9.4 | 59.1 | 38.1 | 46.4 | 52.3 |
| VLM2Vec-V2 (Qwen2-VL-2B) | 62.9 | 56.3 | 69.5 | 77.3 | 64.9 | 39.3 | 34.3 | 28.8 | 38.5 | 34.9 | 75.5 | 44.9 | 79.4 | 39.4 | 65.4 | 58.0 |
| CAFe (LLaVA-OV-7B) | 63.6 | 61.7 | 69.1 | **87.6** | 67.6 | 35.8 | 58.7 | 34.4 | 39.5 | 42.4 | 70.7 | 49.6 | 79.5 | 38.1 | 63.9 | 60.6 |
| DUME (Qwen2-VL-2B) | 59.3 | 55.0 | 66.3 | 78.0 | 62.5 | 37.7 | 46.6 | 17.1 | 30.0 | 33.2 | 67.6 | 43.3 | 47.1 | 33.8 | 52.8 | 52.7 |
| DUME (Qwen2-VL-7B) | 64.2 | 57.0 | 70.8 | 81.8 | 66.4 | 32.9 | 47.4 | 8.6 | 28.0 | 29.4 | 67.1 | 35.2 | 82.6 | 34.9 | 60.3 | 55.9 |
| *Ours* | | | | | | | | | | | | | | | | |
| UME-R1 (Qwen2-VL-2B) | 64.8 | 62.8 | 67.6 | 77.2 | 66.6 | 44.3 | 51.2 | 32.9 | 39.7 | 42.2 | 72.4 | 46.2 | 79.2 | 37.2 | 63.9 | 60.1 |
| UME-R1 (Qwen2-VL-7B) | **67.1** | **69.2** | **71.9** | 84.9 | **71.3** | **48.6** | **60.7** | **38.2** | 39.3 | **47.5** | 75.7 | 50.5 | 83.7 | 37.6 | 67.1 | **64.5** |
| *Oracle* | | | | | | | | | | | | | | | | |
| UME-R1 (Qwen2-VL-2B) | 67.6 | 67.5 | 71.2 | 80.1 | 70.2 | 47.0 | 58.7 | 37.2 | 48.8 | 47.9 | 76.8 | 51.5 | 82.6 | 41.5 | 68.2 | 64.4 |
| △− Ours | +2.8 | +4.7 | +3.6 | +2.9 | +3.6 | +2.7 | +7.5 | +4.3 | +9.1 | +5.7 | +4.4 | +5.3 | +3.4 | +4.3 | +4.3 | +4.3 |
| UME-R1 (Qwen2-VL-7B) | 69.1 | 73.2 | 74.8 | 87.4 | 74.2 | 51.6 | 67.2 | 39.6 | 49.6 | 52.2 | 79.7 | 55.8 | 86.0 | 40.7 | 70.8 | 68.1 |
| △− Ours | +2.0 | +4.0 | +2.9 | +2.5 | +2.9 | +3.0 | +6.5 | +1.4 | +10.3 | +4.7 | +4.0 | +5.3 | +2.3 | +3.1 | +3.7 | +3.6 |

Table 2: Ablation study of the RL stage on images, videos, and visual documents.

| # | Model | Image | Video | VisDoc | ALL |
|---|---|---|---|---|---|
| 1 | UME-R1 (Qwen2VL-2B) | **66.6** | **42.2** | **63.9** | **60.1** |
| 2 | w/o RL (UME) | 65.2 ↓1.4 | 41.2 ↓1.0 | 63.5 ↓0.4 | 59.1 ↓1.0 |
| 3 | w/o similarity gap reward | 65.2 ↓1.4 | 41.2 ↓1.0 | 63.6 ↓0.3 | 59.2 ↓0.9 |
| 4 | w/o ranking reward | 66.0 ↓0.6 | 41.8 ↓0.4 | 63.3 ↓0.6 | 59.6 ↓0.5 |
| 5 | w/ threshold reward | 65.6 ↓1.0 | 41.7 ↓0.5 | 63.5 ↓0.4 | 59.4 ↓0.7 |

embeddings. The results in the Rows 3 and 4 show that jointly considering ranking and similarity differences in the reward is essential. Ranking offers supervision that aligns more closely with downstream tasks, but for relatively easy samples, the ranking reward often saturates. In such cases, similarity differences help guide the model toward learning more effective reasoning paths. In addition, we explore using a fixed threshold (set to 0.5) as the evaluation criterion for assigning rewards, where positive pairs exceeding the threshold receive a reward of 1 and others receive 0. The results in Row 5 show that this approach is mainly beneficial for video tasks but provides limited improvement for other modalities. We attribute this to the varying similarity distributions across task categories, which make it difficult to define a single fixed threshold. Developing an adaptive threshold for reward assignment may be a promising solution.

**Impact of Reasoning-Driven Generative Embedding Training on Discriminative Embeddings.** While UME-R1 is primarily designed for reasoning-driven generative embeddings, it also supports discriminative embeddings. In this study, we investigate how the SFT stage and the RL stage affect the performance of discriminative embeddings. Table 3 reports the performance of 2B-parameter models DUME, UME (without RL training), and UME-R1.

Table 3: Comparison of UME and UME-R1 using only discriminative embeddings against DUME under the same training settings.

| Model | Image | Video | VisDoc | ALL |
|---|---|---|---|---|
| DUME | 62.5 | 33.2 | 52.8 | 52.7 |
| UME | 63.2 ↑0.7 | 34.4 ↑1.2 | 60.3 ↑7.5 | 55.7 ↑3.0 |
| UME-R1 | 64.0 ↑1.5 | 34.4 ↑1.2 | 60.3 ↑7.5 | 56.0 ↑3.3 |

Under the same training settings, introducing reasoning-driven generative embeddings and the next-token prediction objective during the SFT stage improves the

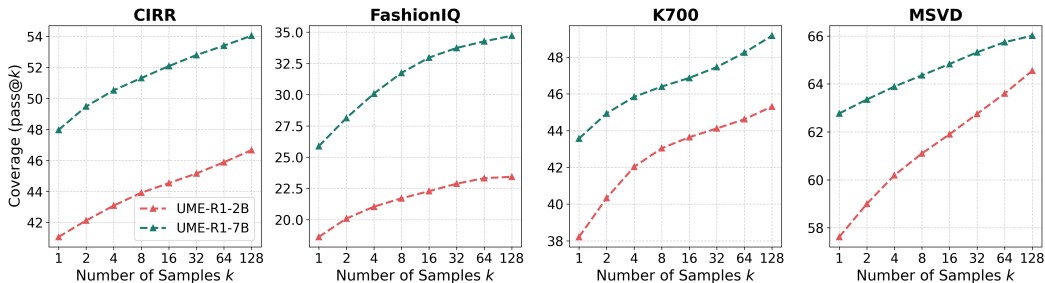

Figure 3: pass@$k$ curves of UME-2B and UME-7B across multiple datasets.

overall score of discriminative embeddings across 78 tasks by 3 points. Notably, for visual document tasks, the improvement reaches 7.5 points, likely due to the limited amount of such data in the training set, suggesting that incorporating the generative embedding and the next-token prediction objective provides richer supervisory signals. Furthermore, UME-R1 achieves an additional 0.4-point improvement over UME in the overall score. Although the RL stage only optimizes the reasoning-driven generative embeddings, it does not compromise the performance of the discriminative embeddings, indicating that the two types of embeddings do not conflict during training.

## 4.4 DEEP ANALYSIS

**Potential of Reasoning-Driven Generative Embeddings for Inference-Time Scaling.** One of the key characteristics of generative reasoning models is their ability to scale at inference time, meaning that performance can be improved by allocating more computing resources. Motivated by this, we explore whether reasoning-driven generative embeddings possess similar potential for inference-time scaling. To this end, we evaluate model coverage (pass@$k$) on four randomly selected test sets from the image and video modalities, each containing 128 randomly sampled examples. Pass@$k$ considers a problem solved if any of the $k$ sampled outputs is correct, thereby indicating the model's ability to retrieve the correct result through multiple attempts. To reduce variance in coverage estimation, we apply the unbiased estimation formula proposed by Brown et al. (2024). As illustrated in Figure 3, both UME-R1-2B and UME-R1-7B yield improved embedding representations through repeated sampling, underscoring that reasoning-driven generative embeddings also hold strong promise for inference-time scaling. Appendix G presents visual illustrations of how repeated sampling affects retrieval results.

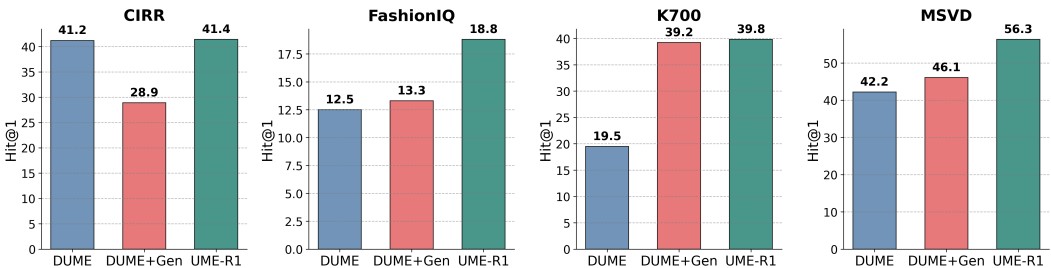

Figure 4: Comparison between DUME, DUME+Gen, and UME-R1. DUME+Gen denotes the approach in which an external model first generates reasoning and summaries, followed by DUME to obtain the corresponding embeddings.

**External-Enhanced Discriminative Embeddings vs. Self-Generated Generative Embeddings.** We further investigate an approach where an external reasoning model generates reasoning and summaries, subsequently encoded by discriminative embedding model to obtain representations. We evaluate whether this approach enhances performance and compare it with our proposed self-generated method. Concretely, we evaluate the 2B model on previously extracted test set, employing the 9B GLM-4.1V-Thinking (Hong et al., 2025a) as the external reasoning model. As shown in Fig-

ure 4, incorporating an external model can enhance discriminative embeddings on certain tasks, with improvements of 19.7 and 3.9 observed on K700 and MSVD, respectively. However, this approach may also degrade performance, exemplified by a 12.3-point drop on CIRR. Importantly, UME-R1 consistently outperforms DUME+Gen, indicating that self-generated reasoning and summaries are more efficient and effective than even a stronger external model for producing high-quality embedding representations.

## 5 RELATED WORK

### 5.1 MULTIMODAL LARGE LANGUAGE MODEL

Multimodal large language models (MLLMs) (OpenAI, 2023; Liu et al., 2023; Chen et al., 2023; Li et al., 2024; Wang et al., 2024) have achieved remarkable progress across a wide range of multimodal understanding tasks (Li et al., 2023; Lan et al., 2025a; Lin et al., 2025b). The emergence of Large Reasoning Models (LRMs), exemplified by GPT-4o (Hurst et al., 2024) and DeepSeek-R1 (Guo et al., 2025), has catalyzed the development of various strategies to elicit chain-of-thought (CoT) reasoning (Wei et al., 2022b; Muennighoff et al., 2025; Zhang et al., 2026) within MLLMs. Among the most prominent is the use of reinforcement learning with verifiable reward signals to enhance visual reasoning (Zhou et al., 2025a; Zhan et al., 2025; Liu et al., 2025b; Shen et al., 2025a). However, to our knowledge, no prior work has applied reinforcement learning with verifiable reward to embedding tasks, primarily due to such tasks are non-generative and do not have definitive answers.

### 5.2 UNIVERSAL MULTIMODAL EMBEDDINGS

Universal multimodal embedding models aim to encode inputs of various modalities into vector representations, facilitating a range of multimodal tasks such as image-text retrieval (Wu et al., 2021; Zhang et al., 2024a), automatic evaluation (Hessel et al., 2021), and retrieval-augmented generation (RAG) (Zhao et al., 2023). Early vision-language models (VLMs) (Radford et al., 2021; Jia et al., 2021; Zhai et al., 2023) primarily used a dual-encoder architecture and were trained with contrastive learning on large-scale image–text datasets. Although these models exhibited strong representational capabilities, they still suffered from deficiencies such as poor understanding of interleaved image–text inputs and a tendency to behave like bag-of-words (Yüksekgönül et al., 2023).

To address these issues, VLM2Vec (Jiang et al., 2025) and MM-Embed (Lin et al., 2025a) convert MLLMs into multimodal embedding models through contrastive learning, leveraging MLLMs' strong multimodal understanding and inherent advantages in handling interleaved image–text inputs. Given the limited scale of existing multimodal embedding datasets, MegaPairs (Zhou et al., 2025c) and GME (Zhang et al., 2025b) introduce automated data synthesis pipelines to generate large-scale pairs, thereby further improving the performance of MLLM-based multimodal embedding models. On the other hand, some works focus on negative sample selection or learning, for example, UniME (Gu et al., 2025a) filters out false negatives and easy negatives during training based on similarity, while LLaVE (Lan et al., 2025b) and QQMM (Xue et al., 2025) estimate negative difficulty and weight negatives accordingly. Furthermore, B3 (Thirukovalluru et al., 2025) introduces a hard negative mining method that leverages community detection to construct training batches enriched with in-batch negatives.

Additionally, some studies explore how to preserve MLLMs' generative strengths when converting them from generative to discriminative models. VladVA (Ouali et al., 2025) and CAFe (Yu et al., 2025a) combine a contrastive objective with autoregressive language modeling to prevent catastrophic forgetting of the models' generative abilities while enhancing their discriminative capabilities. Moreover, Ju & Lee (2025) design hierarchical prompts to elicit powerful discriminative embeddings from generative models in a zero-shot manner. Despite these advances, existing MLLM-based embedding models remain limited to producing discriminative embeddings and therefore do not exploit MLLMs' generative and reasoning capabilities. In contrast, UME-R1 can generate discriminative or reasoning-driven generative embeddings on demand, demonstrating the substantial potential of harnessing MLLMs' reasoning power for embedding tasks.

## 6 CONCLUSION

In this work, we pioneer the exploration of reasoning-driven generative embeddings and propose UME-R1, a universal multimodal embedding framework that unifies discriminative and reasoning-driven generative embeddings. To support this, we construct an SFT dataset by augmenting existing multimodal embedding benchmarks with reasoning and summaries produced by a thinking-capable MLLM. Fine-tuning on this dataset enables the model to produce both embedding types. We further apply reinforcement learning with a reward function that incorporates similarity gaps and ranking, encouraging reasoning trajectories that enhance reasoning-driven generative embeddings. Experiments on MMEB-V2, spanning 78 tasks across video, image, and visual document domains, show that reasoning-driven generative embeddings yield significant gains over discriminative ones. Finally, oracle and inference-time analyses suggest that UME-R1 holds substantial headroom for further improvement.

Our work highlights three promising directions for future research: 1) developing mechanisms that allow the model to adaptively decide whether to produce discriminative or reasoning-driven generative embeddings based on the input; 2) constructing more challenging RL datasets or designing more effective RL training strategies to encourage the model to produce reasoning and summaries that more conducive to embedding quality; and 3) exploring inference-time scaling techniques to further enhance the quality of reasoning-driven generative embeddings. In general, UME-R1 establishes a new direction for reasoning-driven generative multimodal embeddings and lays a foundation for future research.

## ACKNOWLEDGMENTS

The project was supported by National Key R&D Program of China (No. 2022ZD0160501), Natural Science Foundation of Fujian Province of China (No. 2024J011001), and the Open Competition for Innovative Projects of Xiamen (No.3502Z20251012). We also thank the reviewers for their insightful comments.

## ETHICS STATEMENT

This work complies with the ICLR Code of Ethics and does not involve the collection of new human subject data or any personally identifiable information. All datasets used in this study are publicly available and widely adopted in the research community. Additionally, the constructed data in our experiments is derived from existing models and datasets, without introducing any new sensitive, private, or proprietary content. We have carefully ensured that our methodology and experiments comply with relevant ethical standards, including fairness, transparency, and reproducibility.

## REPRODUCIBILITY STATEMENT

To facilitate reproducibility, we will release code, datasets, and trained models used in this work. The code has already been included in the supplementary materials submitted with this paper. Detailed descriptions of the dataset construction, model architectures, and training procedures are provided in both the main text and the appendix. These resources are intended to enable other researchers to reproduce the results reported in this work and build upon our methods.

## USE OF LARGE LANGUAGE MODELS

In the preparation of this paper, we use Large Language Models (LLMs) solely to aid in writing and polishing the text, including improving clarity, grammar, and readability. LLMs are not used for generating scientific content, experimental design, analysis, or conclusions. All technical ideas, experiments, and results reported in this paper are entirely the work of the authors.

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

## A    LIMITATIONS OF UME-R1

Although UME-R1 demonstrates that reasoning-driven generative embeddings exhibit stronger performance and greater potential than discriminative embeddings, they incur higher training and inference costs due to the generation of long CoT and summaries. However, this also opens a new avenue for improving embedding performance beyond scaling model size, namely scaling computation. Moreover, while our oracle upper-bound analysis empirically shows the complementarity between discriminative and reasoning-driven generative embeddings, designing a practical router to select between the two in real-world applications remains an open problem. Finally, there is still room for further performance improvement in our current RL setup, for example, by constructing harder negative examples for RL training or scaling up the training instances.

## B    TRAINING AND INFERENCE COST

In this section, we discuss the training cost of UME-R1 as well as the inference overhead of reasoning-driven generative embeddings compared to discriminative embeddings.

Under the same training configuration, DUME requires 1487 H20 GPU-hours for fine-tuning, whereas UME-R1 incurs 2336 H20 GPU-hours in the SFT stage and 1344 H20 GPU-hours in the RL stage.

Table 4: Comparison of inference speed between discriminative and reasoning-driven generative embeddings across different datasets. The embedding type produced is indicated in parentheses.

| Model | CIRR | FashIQ | K700 | MSVD |
|---|---|---|---|---|
| UME-R1 (Generative) | 1.48 samples/s | 1.14 samples/s | 0.50 samples/s | 1.10 samples/s |
| UME-R1 (Discriminative) | 20.0 samples/s | 19.1 samples/s | 1.59 samples/s | 28.0 samples/s |

As for inference cost, we evaluate inference speed on CIRR, FashionIQ, K700, and MSVD using a single L40s GPU under the vLLM framework. The batch size is set to 8 for image modalities and 4 for video modalities. As shown in Table 4, reasoning-driven generative embeddings indeed introduce a noticeably higher inference overhead, especially when the input token length is short. The speed gap narrows as the input token length increases. Nevertheless, the stronger performance, better interpretability, and the ability to scale computation to further enhance embedding quality make the cost of reasoning-driven generative embeddings well justified.

## C    EXAMPLE OF DATA CONSTRUCTION

The prompt template for SFT CoT annotation is provided as follows:

> **Prompt Template for Reasoning Annotation**
>
> {query/candidate}
> *The above input is a query/candidate for retrieval. Carefully examine and analyze the above input (which may include text, images, videos, or any combination). Identify and describe the key elements present in the input, such as the main topic, important entities, relationships, context, and any notable features or details that contribute to the overall meaning. Finally, synthesize your analysis and reflection into a single word or a concise sentence that best captures the essence of the input for retrieval purposes. If the input is a phrase or word, the summary is that word itself.*

We present examples of our constructed cold-start dataset in Figures 5, 6, and 7 to illustrate the typical query–target pairs it contains. For RL training, we sampled roughly equal numbers of instances from each modality while ensuring a balanced numbers across different datasets within each modality. In particular, for the image modality, pairs were drawn only from OK-VQA (Marino et al., 2019), ChartQA (Masry et al., 2022), CIRR (Liu et al., 2021), A-OKVQA (Marino et al., 2019), and Visual7W (Zhu et al., 2016), as the tasks in the other image datasets are relatively simple.

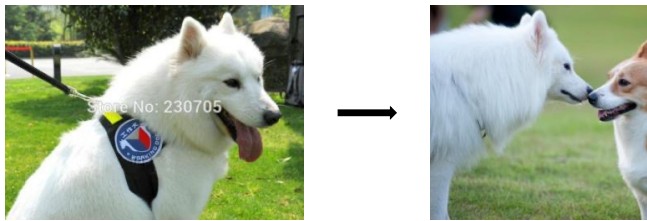

**Query:**
Given an image, find a similar everyday image with the described changes: Shows a similar dog touching noses with a smaller light brown and white dog.*<disc_emb>*
Represent the above input text, images, videos, or any combination of the three as embeddings. First output the thinking process in *<think> </think>* tags and then summarize the entire input in a word or sentence. Finally, use the *<gen_emb>* tag to represent the entire input.
*<think>*So, let\'s analyze the input. The query is asking for a similar everyday image with a dog touching noses with a smaller light brown and white dog. First, identify key elements: main topic is a dog image with interaction (nose touching) with a smaller dog of specific colors (light brown and white). Important entities: dogs, touching noses, smaller light brown and white dog. Relationships: interaction between two dogs, one larger (similar to original dog maybe), one smaller with light brown and white. Context: everyday image, so a normal scene, not a special setting. Notable features: size difference, color of the smaller dog, the action of touching noses. Now, synthesize into a concise sentence for retrieval purposes. Maybe something like "Dog touching noses with smaller light brown and white dog in everyday setting" or a shorter version. Wait, the task says a single word or concise sentence. Let\'s make it concise. "Dog interacting with smaller light brown and white dog by nose touching" or even shorter. Hmm. Let\'s check the original image: the first image is a white dog on a leash. The query is for similar image (so maybe same type setting, park maybe) with two dogs, larger dog (similar to original maybe) touching noses with smaller light brown and white one. So the essence is finding an image of a dog (probably similar to first dog or same context) touching noses with a smaller, light brown and white dog. So the summary could be a sentence that captures the key elements: "Dog touching nose with smaller light brown and white dog." But need to make sure it\'s concise. Alternatively, "Similar image of dog touching noses with smaller light brown and white dog." Wait, the original input is a query for retrieval, so the output should be a concise summary. Let me think again. The input is a query asking for an image of a dog touching noses with a smaller light brown and white dog (similar to an everyday image). So key elements: dog, touch noses, smaller, light brown and white, everyday image. So the summary could be "dog touching nose with smaller light brown and white dog" or a brief description. Since it\'s a retrieval query, maybe "similar dog touching smaller light brown and white dog with noses". But need to be concise. Let\'s go with "dog touching noses with smaller light brown and white dog" as that captures all key elements.*</think><answer>*dog touching noses with smaller light brown and white dog\n*<gen_emb>*
**Target:**
Represent the given image.*<disc_emb>*
Represent the above input text, images, videos, or any combination of the three as embeddings. First output the thinking process in *<think> </think>* tags and then summarize the entire input in a word or sentence. Finally, use the *<gen_emb>* tag to represent the entire input.
*<think>*So, let's analyze the image. There are two dogs touching noses, one white and fluffy, the other tan and white (like a corgi). The setting is a grassy area. The main elements are the two dogs interacting, their breeds/color, and the outdoor environment. So the key elements are the interaction between the two dogs, their appearance, and the grassy background. The essence would be two dogs touching noses in a grassy area.*</think><answer>*Two dogs nose-to-nose on grass\n(Interpreting the image: two dogs, one white fluffy breed and one tan/white corgi-like, touching noses in a grassy setting. Syncretized essence for retrieval: two dogs nose-to-nose on grass.)\n*<gen_emb>*

Figure 5: Example from the constructed cold-start dataset (Case 1). The orange part represents the original data, the blue part denotes the added prompt, the black part indicates the reasoning content, and the green part shows the summary. orange segments correspond to the original data, blue segments represent the added prompts, black segments capture the reasoning process, and green segments provide the summaries.

Source: https://www.industrydocuments.ucsf.edu/docs/fxwg0023

**Query:**

Represent the given image with the following question: What is the Account Name? *<disc_emb>*
Represent the above input text, images, videos, or any combination of the three as embeddings. First output the thinking process in *<think> </think>* tags and then summarize the entire input in a word or sentence. Finally, use the *<gen_emb>* tag to represent the entire input.

*<think>*So, let\'s look at the image. The question is asking for the Account Name. In the document, under "Account" it has "Janeto\'s Super Market" written. So we need to extract that. The key element here is the account name, which is clearly labeled next to "Account".*</think><answer>*Janeto\'s Super Market\n*<gen_emb>*

**Target:**

Janeto's Super Market*<disc_emb>*
Represent the above input text, images, videos, or any combination of the three as embeddings. First output the thinking process in *<think> </think>* tags and then summarize the entire input in a word or sentence. Finally, use the *<gen_emb>* tag to represent the entire input.

*<think>*So, let\'s analyze the input. The input says "Janeto\'s Super Market" as a candidate for retrieval. I need to identify key elements:\n\nMain topic: Janeto\'s Super Market (which is likely a store name). Important entities: "Janeto\'s" ( Likely a business name, super market as the type). Relationships: It\'s a business (super market) owned or named "Janeto\'s". Context: This seems to be a name for a supermarket business. Notable features: It\'s presented as a candidate for retrieval, so it\'s a name that might be used for identification.\n\nSynthesize this into a single word or concise sentence. The main essence is a supermarket business named Janeto\'s. So the summary would be the business name, or a concise description. But since the task says "single word or concise sentence", "Janeto\'s Super Market" is the key, but maybe shorter. Alternatively, "Janeto\'s Supermarket" as the essence. Wait, the input is "Janeto\'s Super Market" (with an apostrophe), so the essence is the supermarket with that name.*</think><answer>*Janeto\'s Super Market (essence: supermarket named Janeto\'s)\n(Concise summary for retrieval: "Janeto\'s super market" or "Janeto\'s Super Market") \n*<gen_emb>*

Figure 6: Example from the constructed cold-start dataset (Case 2).

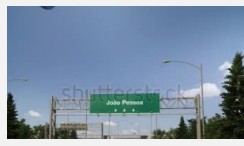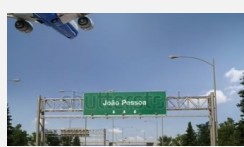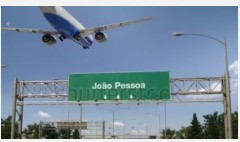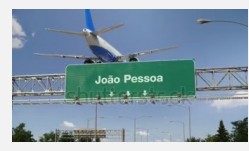

**Query:**

Find a video that contains the following visual content: This video captures the heart-pounding moment of an airplane, with a visibly blue and white body probably belonging to a commercial airline, flying at a low altitude over an overhead green sign. The sign reads "João Pessoa" in white capital letters, it appears to be naming a city or a specific area. It\'s made of reflective material allowing high visibility even in darker conditions. This particular sign is prominently installed on metal pillars with a series of lights to illuminate it, suggesting it is likely situated near a busy motorway or highway. As the airplane ascends, its undercarriage is still visible, highlighting that it has likely just taken off from a nearby airport. We do not have clear visibility of the airplane\'s livery or any markings that could identify the specific airline or model of the plane. The sky is clear, with scarce, fluffy clouds scattered, suggesting good weather conditions. No other objects or subjects are visible in the video other than the stationary overhead sign and the airplane making its ascent.*<disc_emb>*

Represent the above input text, images, videos, or any combination of the three as embeddings. First output the thinking process in *<think> </think>* tags and then summarize the entire input in a word or sentence. Finally, use the *<gen_emb>* tag to represent the entire input.

*<think>*So, let\'s analyze the input. The user is looking for a specific video with a detailed description of the visual content. Let\'s break down the key elements:\n\nMain topic: A video capturing an airplane (commercial, blue and white) flying low over a "João Pessoa" sign. \n\nImportant entities: \n- Airplane: blue/white, low altitude takeoff, undercarriage visible, livery/terms unclear.\n- Sign: "João Pessoa" (white letters, green background, reflective, on metal pillars with lights), near motorway/highway.\n- Context: clear sky, low cloud, no other objects.\n\nRelationships: Airplane is flying over the sign which is near a motorway. The sign\'s purpose is to name a city/area, the airplane is ascending (justice takeoff). \n\nNotable features: Color of airplane, visibility of sign, sky conditions, absence of other objects.\n\nSynthesize into a concise summary. The essence is video of commercial airplane flying low over "João Pessoa" sign near motorway with clear sky.*</think><answer>*Commercial airplane flies low over "João Pessoa" sign near motorway, clear sky.\n<gen_emb>

**Target:**

Understand the content of the provided video.*<disc_emb>*

Represent the above input text, images, videos, or any combination of the three as embeddings. First output the thinking process in *<think> </think>* tags and then summarize the entire input in a word or sentence. Finally, use the *<gen_emb>* tag to represent the entire input.

*<think>*Got it, let\'s analyze the input. The user provided multiple images (though text describes them as video frames maybe) showing green highway signs with "João Pessoa" and airplanes in some. Let\'s identify key elements: green highway signs, João Pessoa as a place name, airplanes (aviation elements), the signs\' structure (metal frames), sky, trees, etc. The main topic is highway signs for João Pessoa with airplanes (maybe an airport area). Important entities: "João Pessoa" signs, airplanes, highway infrastructure. Relationships: signs are for directing to João Pessoa, airplanes indicate proximity to an airport. Context: outdoor setting, clear sky, urban/airport infrastructure. Notable features: repeated green sign with João Pessoa, airplanes flying above signs. \n\nSynthesize into a single word or concise sentence. The core is highway signs for João Pessoa with airplanes, so maybe "João Pessoa highway signs with airplanes" but make it concise. Or "João Pessoa highway signage with aircraft". But need to capture essence. Alternatively, "João Pessoa highway signs and airplanes*".</think><answer>*João Pessoa highway signs with aircraft\n(Condensed to capture the key elements: the main subject "João Pessoa" on highway signs plus the included aircraft.)\n<gen_emb>

Figure 7: Example from the constructed cold-start dataset (Case 3).

# D   DETAILED SCORES OF MMEB-V2

Table 5: The detailed results of the baselines and UME-R1 the full MMEB-v2 benchmark. We only include the best version of each series of previous models in the table. Numbers in parentheses represent the task count for each category.

| | ColPali v1.3 | GME-7B | LamRA-Qwen2.5-VL | VLM2Vec-7B | VLM2Vec-V2.0 | CAFe-7B | DUME-2B | DUME-7B | UME-R1-2B | UME-R1-7B |
|---|---|---|---|---|---|---|---|---|---|---|
| Avg - All (78 tasks) | 44.4 | 57.8 | 47.4 | 52.3 | 58.0 | 60.6 | 52.7 | 55.9 | 60.1 | 64.5 |
| Avg - Image (36 tasks, Hit@1) | 34.9 | 56.0 | 52.4 | 65.5 | 64.9 | 67.6 | 62.5 | 66.4 | 66.6 | 71.3 |
| Avg - Video (18 tasks, Hit@1) | 28.2 | 38.4 | 33.6 | 33.7 | 34.6 | 42.4 | 33.2 | 29.4 | 42.2 | 47.5 |
| Avg - Visdoc (24 tasks, NDCG@5) | 71.0 | 75.2 | 50.2 | 46.4 | 65.4 | 63.9 | 52.8 | 60.3 | 63.9 | 67.1 |
| I-CLS (10) | 40.3 | 57.7 | 51.7 | 62.7 | 62.9 | 63.6 | 59.3 | 64.2 | 64.8 | 67.1 |
| I-QA (10) | 11.5 | 34.7 | 34.1 | 56.9 | 56.3 | 61.7 | 54.9 | 57.0 | 62.8 | 69.2 |
| I-RET (12) | 48.1 | 71.2 | 66.9 | 69.4 | 69.5 | 69.1 | 66.3 | 70.8 | 67.6 | 71.9 |
| I-VG (4) | 40.3 | 59.3 | 56.7 | 82.2 | 77.3 | 87.6 | 78.0 | 81.8 | 77.2 | 84.9 |
| V-CLS (5) | 26.7 | 37.4 | 32.9 | 39.1 | 39.3 | 35.8 | 37.7 | 32.9 | 44.3 | 48.6 |
| V-QA (5) | 37.8 | 50.4 | 42.6 | 30.0 | 34.3 | 58.7 | 46.6 | 47.4 | 51.0 | 60.7 |
| V-RET (5) | 21.6 | 28.4 | 23.2 | 29.0 | 28.8 | 34.4 | 17.1 | 8.6 | 32.9 | 38.2 |
| V-MR (3) | 25.5 | 37.0 | 37.2 | 38.9 | 36.8 | 39.5 | 30.0 | 28.0 | 39.7 | 39.3 |
| VD-Vidore-V1 (10) | 83.6 | 89.4 | 56.3 | 56.9 | 75.7 | 70.7 | 67.6 | 67.1 | 72.4 | 75.7 |
| VD-Vidore-V2 (4) | 52.0 | 55.6 | 33.3 | 9.4 | 45.1 | 49.6 | 43.3 | 35.2 | 46.2 | 50.5 |
| VD-VisRAG (6) | 81.1 | 85.0 | 58.2 | 59.1 | 79.6 | 79.5 | 47.1 | 82.6 | 79.2 | 83.7 |
| VD-OOD (4) | 43.1 | 44.4 | 40.1 | 38.1 | 39.6 | 38.1 | 33.8 | 34.9 | 37.2 | 37.6 |
| ImageNet-1K | 42.4 | 64.6 | 58.9 | 80.1 | 80.8 | 77.3 | 74.6 | 76.6 | 75.3 | 80.4 |
| N24News | 25.5 | 50.5 | 29.8 | 79.7 | 72.9 | 83.2 | 69.7 | 77.2 | 81.1 | 82.3 |
| HatefulMemes | 50.6 | 53.6 | 51.3 | 69.7 | 56.3 | 78.7 | 65.3 | 79.6 | 75.2 | 79.0 |
| VOC2007 | 69.8 | 80.3 | 78.7 | 80.7 | 85.0 | 89.8 | 68.9 | 85.5 | 80.0 | 90.8 |
| SUN397 | 56.1 | 69.5 | 66.5 | 77.4 | 71.0 | 79.9 | 71.4 | 74.6 | 79.4 | 80.3 |
| Place365 | 27.5 | 39.1 | 37.4 | 37.4 | 35.9 | 45.0 | 41.0 | 41.9 | 42.6 | 46.8 |
| ImageNet-A | 14.9 | 41.2 | 36.3 | 58.1 | 47.4 | 55.2 | 41.3 | 48.6 | 50.4 | 53.9 |
| ImageNet-R | 64.6 | 83.9 | 77.0 | 73.9 | 89.3 | 88.0 | 90.7 | 88.8 | 88.7 | 90.1 |
| ObjectNet | 45.6 | 69.0 | 59.4 | 40.1 | 65.2 | 22.5 | 46.2 | 44.8 | 52.0 | 42.3 |
| Country211 | 6.0 | 24.8 | 21.7 | 29.8 | 25.2 | 16.7 | 23.9 | 24.7 | 23.4 | 25.0 |
| OK-VQA | 9.4 | 33.2 | 39.9 | 56.8 | 51.5 | 67.3 | 56.8 | 61.6 | 62.4 | 71.7 |
| A-OKVQA | 6.6 | 21.0 | 34.1 | 47.3 | 43.6 | 63.8 | 46.9 | 51.4 | 51.1 | 58.7 |
| DocVQA | 11.3 | 41.4 | 37.1 | 89.7 | 90.1 | 79.2 | 86.0 | 86.3 | 92.2 | 93.8 |
| InfographicsVQA | 5.0 | 20.3 | 23.7 | 60.0 | 58.8 | 53.3 | 59.2 | 62.3 | 67.7 | 79.2 |
| ChartQA | 5.7 | 17.8 | 15.0 | 56.9 | 47.4 | 48.8 | 39.1 | 49.8 | 64.9 | 75.1 |
| Visual7W | 6.1 | 22.2 | 24.6 | 52.7 | 52.9 | 52.5 | 46.9 | 52.1 | 54.1 | 55.2 |
| ScienceQA | 16.3 | 28.0 | 31.3 | 38.5 | 38.2 | 65.4 | 38.7 | 45.5 | 42.7 | 53.7 |
| VizWiz | 27.6 | 39.0 | 32.0 | 39.9 | 43.3 | 43.8 | 42.0 | 44.3 | 46.8 | 51.6 |
| GQA | 8.3 | 76.9 | 57.4 | 55.1 | 64.9 | 65.7 | 60.2 | 46.9 | 67.3 | 69.3 |
| TextVQA | 18.8 | 46.8 | 46.1 | 71.6 | 72.2 | 76.8 | 73.9 | 69.9 | 78.6 | 83.5 |
| VisDial | 41.2 | 60.8 | 62.5 | 81.9 | 82.7 | 82.7 | 75.9 | 75.7 | 76.6 | 80.7 |
| CIRR | 8.2 | 54.9 | 44.7 | 51.1 | 57.5 | 60.4 | 52.0 | 51.6 | 53.7 | 55.3 |
| VisualNews_t2i | 50.1 | 79.7 | 70.1 | 80.5 | 74.5 | 69.5 | 71.2 | 76.9 | 71.7 | 76.8 |
| VisualNews_i2t | 47.6 | 83.6 | 74.2 | 81.2 | 78.2 | 79.4 | 72.5 | 82.3 | 74.2 | 82.0 |
| MSCOCO_t2i | 59.2 | 71.2 | 65.7 | 77.2 | 75.3 | 75.4 | 74.5 | 77.1 | 75.1 | 78.3 |
| MSCOCO_i2t | 49.9 | 57.7 | 71.1 | 73.9 | 71.4 | 73.1 | 68.3 | 71.2 | 68.9 | 71.4 |
| NIGHTS | 65.5 | 67.6 | 64.4 | 67.6 | 68.6 | 66.7 | 67.5 | 69.6 | 67.2 | 68.1 |
| WebQA | 53.8 | 91.4 | 85.7 | 88.3 | 90.6 | 89.3 | 90.2 | 90.3 | 90.0 | 90.9 |
| FashionIQ | 5.9 | 37.8 | 33.4 | 17.1 | 19.5 | 39.0 | 11.5 | 20.5 | 17.1 | 23.4 |
| Wiki-SS-NQ | 80.5 | 78.2 | 67.0 | 62.3 | 66.9 | 61.2 | 60.0 | 70.6 | 62.0 | 72.5 |
| OVEN | 50.0 | 75.1 | 84.8 | 66.5 | 64.3 | 60.8 | 65.2 | 70.5 | 66.9 | 71.4 |
| EDIS | 64.7 | 96.0 | 78.7 | 85.7 | 84.1 | 71.3 | 86.5 | 92.8 | 88.0 | 92.0 |
| MSCOCO | 36.7 | 31.4 | 36.0 | 75.7 | 67.1 | 84.7 | 68.1 | 72.3 | 69.5 | 72.7 |
| RefCOCO | 64.5 | 60.9 | 57.1 | 87.6 | 87.1 | 89.4 | 85.1 | 86.8 | 83.3 | 91.4 |
| RefCOCO-Matching | 3.9 | 78.4 | 82.6 | 84.6 | 85.8 | 83.0 | 89.3 | 85.1 | 84.4 | 91.1 |
| Visual7W-Pointing | 56.1 | 66.5 | 51.2 | 81.0 | 69.2 | 93.2 | 69.5 | 83.1 | 71.5 | 84.2 |
| K700 | 23.4 | 39.7 | 32.1 | 35.5 | 38.0 | 40.1 | 22.7 | 27.3 | 35.8 | 42.8 |
| SmthSmthV2 | 25.1 | 30.6 | 25.3 | 32.1 | 42.8 | 35.8 | 37.7 | 25.1 | 44.1 | 50.4 |
| HMDB51 | 24.8 | 47.9 | 33.8 | 42.2 | 40.9 | 46.9 | 53.4 | 42.6 | 54.4 | 58.3 |
| UCF101 | 49.4 | 54.7 | 53.0 | 61.8 | 60.0 | 39.6 | 55.7 | 48.8 | 67.2 | 70.0 |
| Breakfast | 10.9 | 14.3 | 20.1 | 23.8 | 14.8 | 16.6 | 18.9 | 20.8 | 20.1 | 21.5 |
| MVBench | 33.7 | 46.6 | 37.6 | 28.5 | 33.7 | 48.9 | 48.8 | 47.4 | 49.9 | 58.2 |
| Video-MME | 30.6 | 39.2 | 35.1 | 27.8 | 30.7 | 46.0 | 39.2 | 40.2 | 41.7 | 47.3 |
| NExTQA | 35.2 | 53.6 | 44.9 | 20.3 | 20.9 | 62.4 | 55.2 | 48.6 | 59.9 | 69.6 |
| EgoSchema | 38.4 | 46.8 | 47.0 | 21.8 | 34.0 | 60.0 | 23.2 | 50.4 | 45.4 | 52.4 |
| ActivityNetQA | 51.3 | 65.6 | 48.5 | 51.4 | 52.3 | 76.0 | 66.7 | 50.2 | 57.8 | 76.0 |
| DiDeMo | 22.8 | 26.4 | 22.8 | 29.3 | 30.4 | 37.8 | 16.9 | 0.10 | 32.4 | 40.0 |
| MSR-VTT | 17.6 | 31.8 | 25.0 | 34.5 | 28.3 | 36.5 | 16.2 | 0.10 | 34.3 | 38.9 |
| MSVD | 45.4 | 49.7 | 41.9 | 46.7 | 48.1 | 56.4 | 34.9 | 28.8 | 55.4 | 60.8 |
| VATEX | 16.7 | 24.9 | 18.7 | 25.5 | 26.5 | 32.0 | 11.1 | 13.8 | 29.9 | 32.6 |
| YouCook2 | 5.3 | 9.1 | 7.5 | 9.0 | 10.6 | 9.5 | 0.06 | 0.00 | 12.7 | 18.5 |
| QVHighlight | 19.9 | 59.5 | 60.9 | 57.7 | 49.4 | 58.4 | 40.3 | 29.4 | 57.5 | 54.9 |
| Charades-STA | 29.0 | 14.0 | 18.8 | 19.8 | 20.2 | 18.7 | 16.1 | 15.8 | 20.4 | 21.9 |
| MomentSeeker | 27.6 | 37.4 | 31.8 | 39.3 | 40.8 | 41.4 | 33.7 | 38.8 | 41.2 | 41.1 |
| ViDoRe_arxivqa | 81.7 | 86.9 | 53.0 | 60.2 | 80.6 | 73.3 | 68.7 | 66.6 | 73.9 | 73.6 |
| ViDoRe_docvqa | 56.6 | 57.5 | 25.4 | 34.7 | 44.9 | 38.3 | 33.6 | 35.8 | 37.9 | 41.1 |
| ViDoRe_infovqa | 84.9 | 91.6 | 72.3 | 70.4 | 83.7 | 80.6 | 74.5 | 72.8 | 76.2 | 80.8 |
| ViDoRe_tabfquad | 86.9 | 94.6 | 66.1 | 78.2 | 89.2 | 80.7 | 78.3 | 89.2 | 86.1 | 90.2 |
| ViDoRe_tatdqa | 70.9 | 74.1 | 25.9 | 27.6 | 43.8 | 37.8 | 35.3 | 38.5 | 40.6 | 46.7 |
| ViDoRe_shiftproject | 75.1 | 96.8 | 27.3 | 38.6 | 60.8 | 52.0 | 61.8 | 61.9 | 66.8 | 65.0 |
| ViDoRe_artificial_intelligence | 95.7 | 99.6 | 72.0 | 67.7 | 88.5 | 86.0 | 74.3 | 69.3 | 85.9 | 89.5 |
| ViDoRe_energy | 94.7 | 95.3 | 65.2 | 60.4 | 86.5 | 84.8 | 78.4 | 68.4 | 83.3 | 85.7 |
| ViDoRe_government_reports | 93.6 | 98.8 | 72.2 | 61.8 | 85.0 | 85.0 | 83.0 | 83.1 | 82.6 | 89.8 |
| ViDoRe_healthcare_industry | 95.9 | 99.3 | 83.8 | 69.9 | 92.2 | 88.4 | 88.2 | 84.9 | 90.8 | 94.3 |
| ViDoRe_esg_reports_human_labeled_v2 | 51.3 | 63.4 | 33.0 | 6.8 | 45.6 | 50.7 | 48.0 | 40.4 | 50.2 | 50.4 |
| ViDoRe_biomedical_lectures_v2_multilingual | 54.7 | 49.5 | 35.9 | 5.1 | 44.3 | 50.9 | 39.8 | 37.4 | 46.2 | 50.7 |
| ViDoRe_economics_reports_v2_multilingual | 49.0 | 54.2 | 31.9 | 13.9 | 43.0 | 54.3 | 44.1 | 29.6 | 45.7 | 57.8 |
| ViDoRe_esg_reports_v2_multilingual | 52.9 | 55.4 | 32.5 | 11.9 | 46.6 | 42.3 | 41.1 | 33.5 | 42.7 | 43.2 |
| VisRAG_ArxivQA | 80.9 | 87.4 | 37.7 | 52.6 | 76.9 | 74.0 | 35.8 | 77.3 | 74.3 | 80.5 |
| VisRAG_ChartQA | 72.3 | 86.1 | 68.2 | 57.7 | 83.7 | 82.7 | 47.2 | 83.4 | 86.0 | 85.0 |
| VisRAG_MP-DocVQA | 82.0 | 89.7 | 72.0 | 60.6 | 88.1 | 75.1 | 35.3 | 83.8 | 75.6 | 83.4 |
| VisRAG_SlideVQA | 85.1 | 92.6 | 71.1 | 54.7 | 84.1 | 87.6 | 61.3 | 91.5 | 87.1 | 91.5 |
| VisRAG_InfoVQA | 83.5 | 88.6 | 67.9 | 66.0 | 82.3 | 87.9 | 64.7 | 88.2 | 84.4 | 89.2 |
| VisRAG_PlotQA | 79.3 | 76.5 | 56.4 | 62.7 | 75.9 | 69.4 | 38.5 | 71.3 | 68.0 | 72.7 |
| ViDoSeek-page | 38.1 | 32.6 | 10.7 | 16.3 | 29.1 | 22.5 | 20.0 | 20.2 | 21.2 | 21.3 |
| ViDoSeek-doc | 87.5 | 90.3 | 63.9 | 69.4 | 79.0 | 73.8 | 69.5 | 73.2 | 75.9 | 75.3 |
| MMLongBench-page | 27.1 | 36.9 | 0.5 | 0.4 | 15.8 | 13.3 | 10.4 | 10.3 | 11.9 | 12.3 |
| MMLongBench-doc | 80.4 | 85.2 | 51.4 | 28.8 | 63.0 | 42.6 | 35.4 | 36.0 | 39.7 | 41.3 |

# E MMEB-V1 BENCHMARK SCORES

Since MMEB-V1 has been widely adopted in prior work, in this section we also report the performance of UME-R1 alongside other baseline models on MMEB-V1. The results in Table 6 demonstrate that UME-R1 achieves the best overall score among models of the same size.

Table 6: Results on the MMEB-V1 benchmark, which comprises a total of 36 image embedding tasks. IND represents the in-distribution dataset, and OOD represents the out-of-distribution dataset. In UniIR, the FF and SF subscripts under CLIP or BLIP represent feature-level fusion and score-level fusion, respectively. CAFe-V1 indicates that the model is trained solely on the MMEB-V1 training data (contains only image data), whereas CAFe-V2 denotes that the model is trained on the MMEB-V2 training data. The best results are marked in bold, and the second-best results are underlined.

| Model | Per Meta-Task Score | | | | Average Score | | |
|---|---|---|---|---|---|---|---|
| | Classification | VQA | Retrieval | Grounding | IND | OOD | Overall |
| **# of Datasets** | 10 | 10 | 12 | 4 | 20 | 16 | 36 |
| *Baseline Models* | | | | | | | |
| CLIP (Radford et al., 2021) | 42.8 | 9.1 | 53.0 | 51.8 | 37.1 | 38.7 | 37.8 |
| BLIP2 (Li et al., 2023) | 27.0 | 4.2 | 33.9 | 47.0 | 25.3 | 25.1 | 25.2 |
| SigLIP (Zhai et al., 2023) | 40.3 | 8.4 | 31.6 | 59.5 | 32.3 | 38.0 | 34.8 |
| OpenCLIP (Cherti et al., 2023) | 47.8 | 10.9 | 52.3 | 53.3 | 39.3 | 40.2 | 39.7 |
| UniIR (BLIP$_{FF}$) (Wei et al., 2024) | 42.1 | 15.0 | 60.1 | 62.2 | 44.7 | 40.4 | 42.8 |
| UniIR (CLIP$_{SF}$) (Wei et al., 2024) | 44.3 | 16.2 | 61.8 | 65.3 | 47.1 | 41.7 | 44.7 |
| Magiclens (Zhang et al., 2024b) | 38.8 | 8.3 | 35.4 | 26.0 | 31.0 | 23.7 | 27.8 |
| *MLLM-based Baseline Models* | | | | | | | |
| E5-V (Jiang et al., 2024) | 21.8 | 4.9 | 11.5 | 19.0 | 14.9 | 11.5 | 13.3 |
| VLM2Vec (Qwen2-VL-2B) (Jiang et al., 2025) | 59.0 | 49.4 | 65.4 | 73.4 | 66.0 | 52.6 | 60.1 |
| VLM2Vec (Qwen2-VL-7B) (Jiang et al., 2025) | 62.6 | 57.8 | 69.9 | 81.7 | 72.2 | 57.8 | 65.8 |
| VLM2Vec-V2 (Qwen2-VL-7B) (Jiang et al., 2025) | 62.9 | 56.3 | 69.5 | 77.3 | 68.8 | 59.9 | 64.9 |
| MMRet-7B (Zhou et al., 2025b) | 56.0 | 57.4 | 69.9 | 83.6 | 68.0 | 59.1 | 64.1 |
| CAFe-V1-7B (Yu et al., 2025a) | 65.2 | 65.6 | 70.0 | 91.2 | 75.8 | 62.4 | 69.8 |
| CAFe-V2-7B (Yu et al., 2025a) | 63.6 | 61.7 | 69.1 | 87.6 | 72.8 | 61.1 | 67.6 |
| mmE5-11B (Chen et al., 2025a) | **67.6** | 62.8 | 70.9 | 89.7 | 72.3 | **66.7** | 69.8 |
| LLaVE-2B (Lan et al., 2025b) | 62.1 | 60.2 | 65.2 | 84.9 | 69.4 | 59.8 | 65.2 |
| LLaVE-7B (Lan et al., 2025b) | 65.7 | 65.4 | 70.9 | **91.9** | 75.0 | 64.4 | 70.3 |
| UniME-4B (Gu et al., 2025b) | 54.8 | 55.9 | 64.5 | 81.8 | 68.2 | 52.7 | 64.2 |
| UniME-7B (Gu et al., 2025b) | 66.8 | 66.6 | 70.6 | 90.9 | 74.6 | 65.8 | 70.7 |
| *Ours* | | | | | | | |
| UME-R1-2B | 64.8 | 62.8 | 67.6 | 77.2 | 71.5 | 60.4 | 66.6 |
| UME-R1-7B | 67.1 | **69.2** | **71.9** | 84.9 | **76.1** | 65.1 | **71.3** |
| *Oracle* | | | | | | | |
| UME-R1-2B | 67.6 | 67.5 | 71.2 | 80.1 | 75.3 | 63.8 | 70.2 |
| UME-R1-7B | 69.1 | 73.2 | 74.8 | 87.4 | 79.2 | 67.9 | 74.2 |

## F COMPARATIVE EXAMPLES OF REASONING-DRIVEN GENERATIVE AND DISCRIMINATIVE EMBEDDINGS

Figures 8, 9, 10, and 11 present several comparative examples of reasoning-driven generative and discriminative embeddings. It can be observed that reasoning-driven generative embeddings are capable of producing effective reasoning and summaries, thereby facilitating the generation of higher-quality embeddings. For example, as shown in Figure 8, UME-R1 first engages in reasoning. The intermediate thought process includes: *"Wait, I think the key is that the food in question is the hot dog ... so the name of the food not white is the hot dog."* As a result, the final summary directly produces *"hot dog"*, yielding a higher-quality embedding and enabling correct retrieval results.

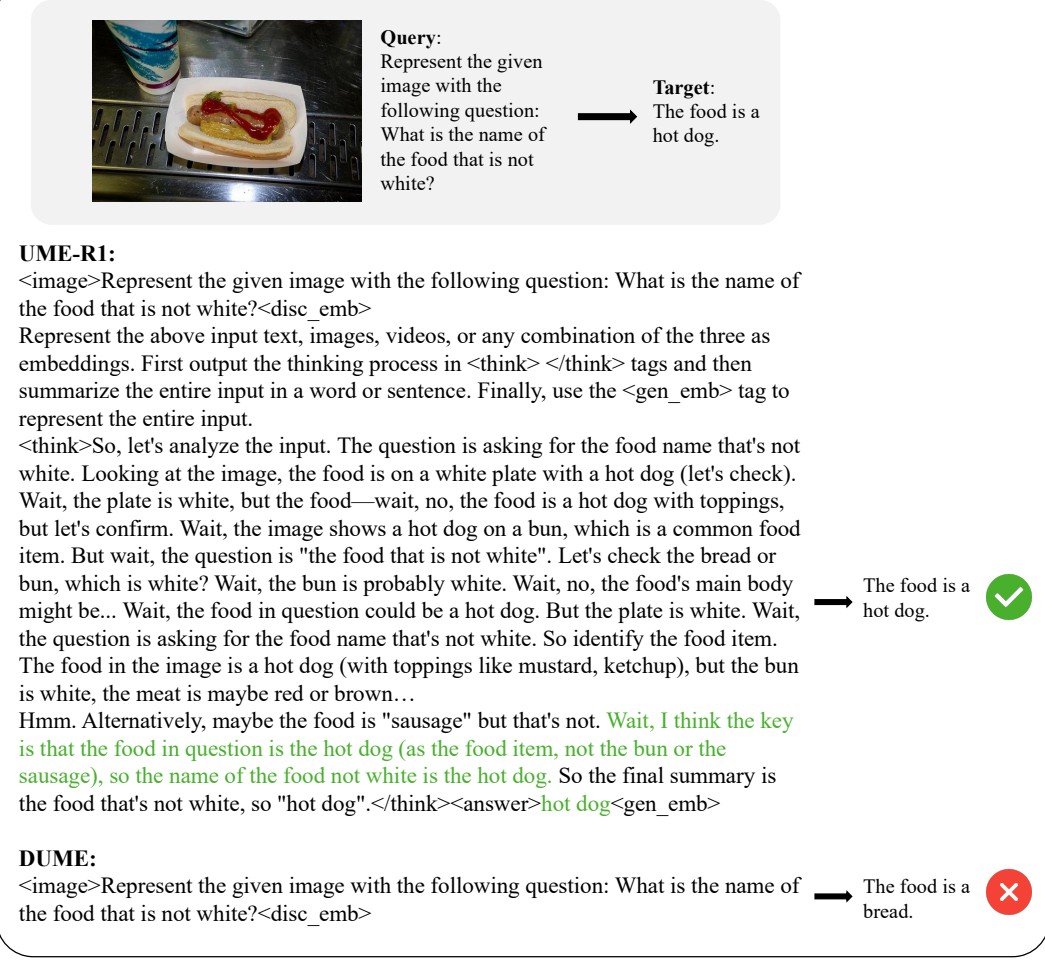

Figure 8: A comparison of reasoning-driven generative and discriminative embeddings is shown (Case 1). Green highlights denote content that positively impacts retrieval performance. UME-R1 demonstrates the results of reasoning-driven generative embeddings, while DUME is trained using standard discriminative contrastive learning. In this example, UME-R1 produces "hot dog" directly during both the reasoning and summarization steps, which helps generate a higher-quality embedding.

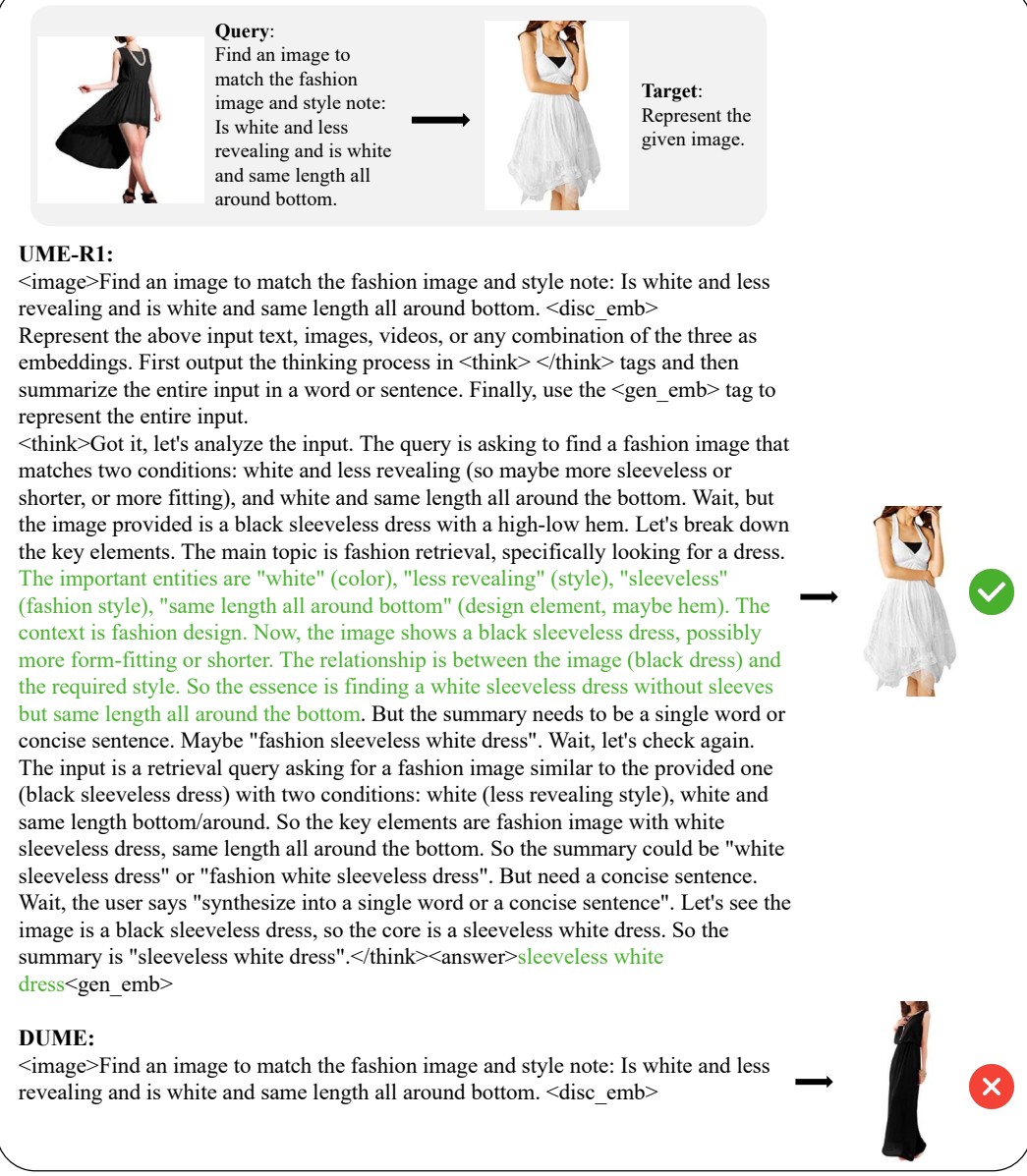

Figure 9: A comparison of reasoning-driven generative and discriminative embeddings is shown (Case 2). In this example, UME-R1 reasons and summarizes that it needs to find a white sleeveless dress with a skirt length matching the provided image, whereas DUME retrieves results based only on the input image and instruction, which do not fully satisfy the requirements.

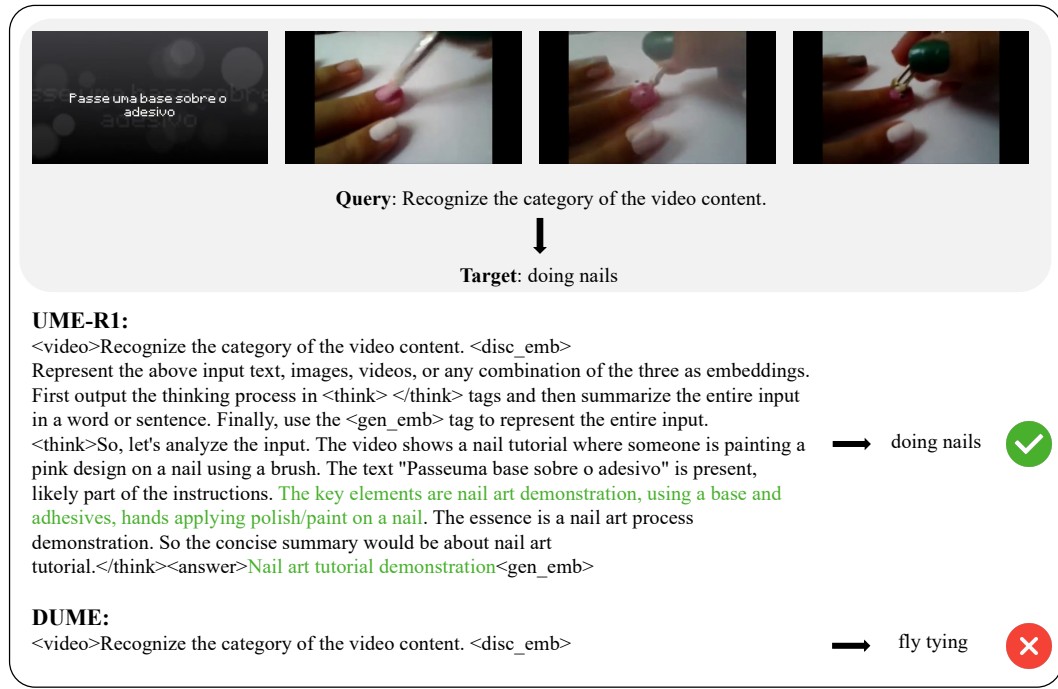

Figure 10: A comparison of reasoning-driven generative and discriminative embeddings is shown (Case 3). In this example, UME-R1 summarizes the video as a "Nail art tutorial demonstration," providing crucial information for correct retrieval.

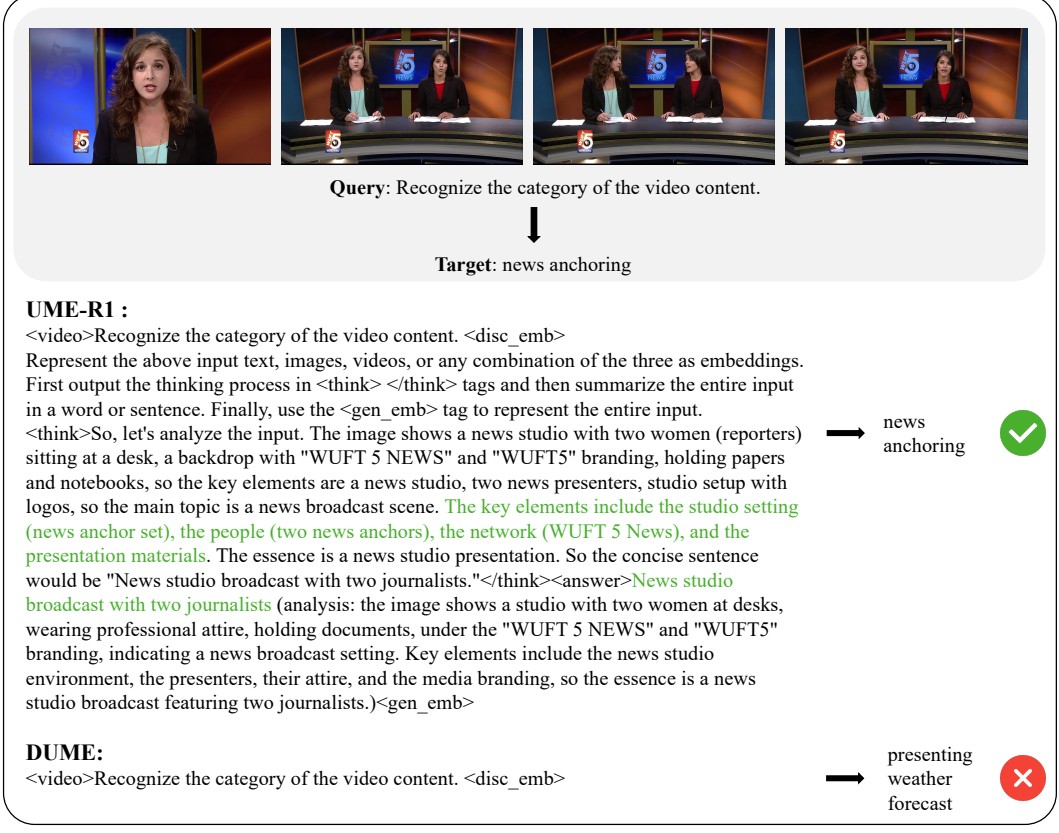

Figure 11: A comparison of reasoning-driven generative and discriminative embeddings is shown (Case 4). In this example, UME-R1 identifies key elements such as "WUFT 5 News" during reasoning, preventing an incorrect retrieval as a weather forecast.

# G    EXAMPLE OF REPEATED SAMPLING

Figures 12, 13, and 14 illustrate the impact of different samplings on the reasoning and summarization of reasoning-driven generative embeddings. Interestingly, we observe that for correctly retrieved examples, the generated reasoning and summaries tend to be more effective. This suggests that reasoning-driven generative embeddings offer a more interpretable approach, potentially allowing the quality of embeddings to be assessed through the generated reasoning and summaries.

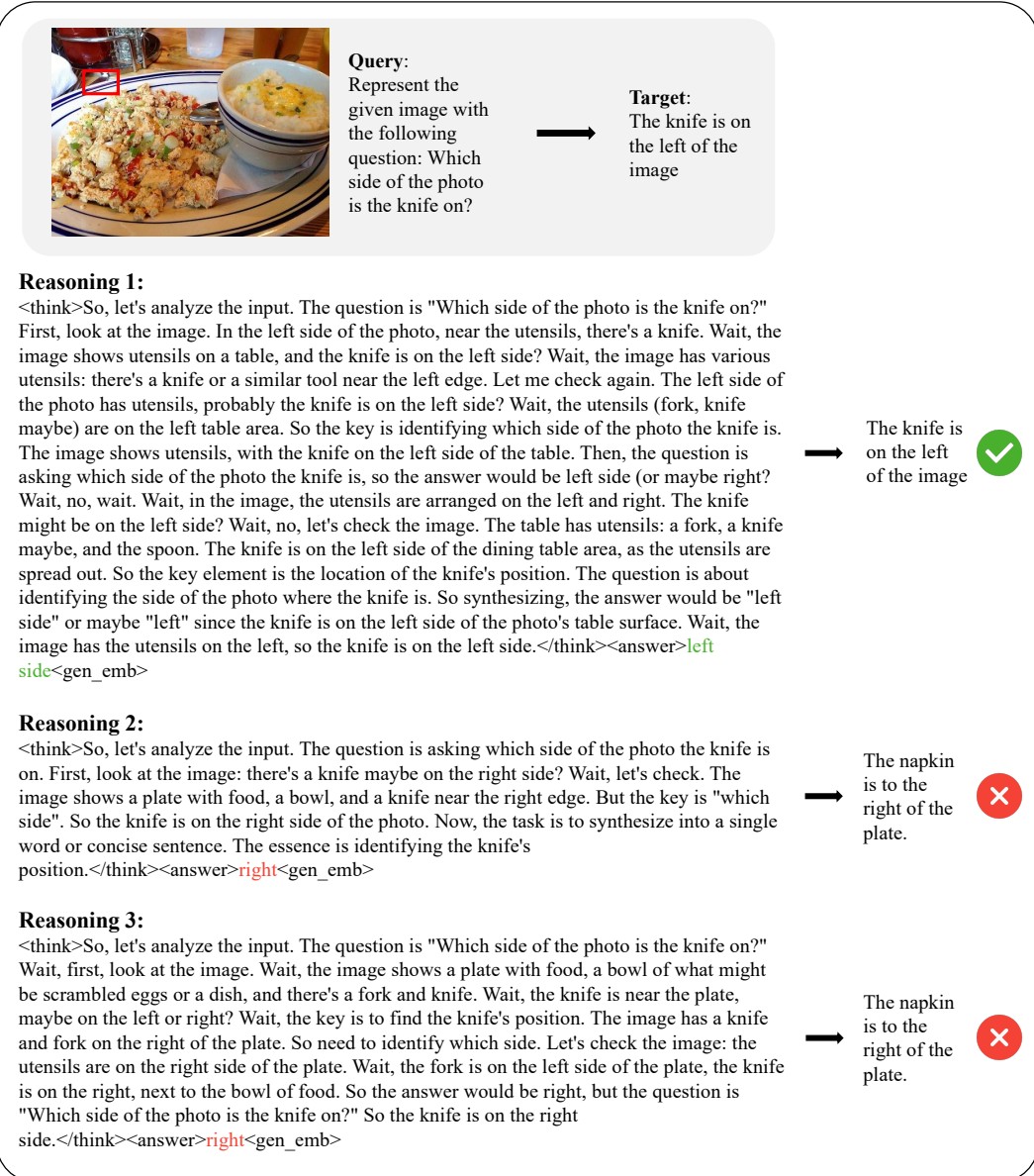

Figure 12: An example showing how repeated sampling leads to variations in model-generated reasoning and summaries, resulting in different retrieval outcomes (Case 1). The green segments indicate correct reasoning or summaries, while the red segments highlight incorrect ones. It can be observed that in correctly retrieved examples, the summaries are more accurate. For instance, when the summary is "left side," the retrieval results include "left of," whereas when the summary is "right," the retrieval results include "right of."

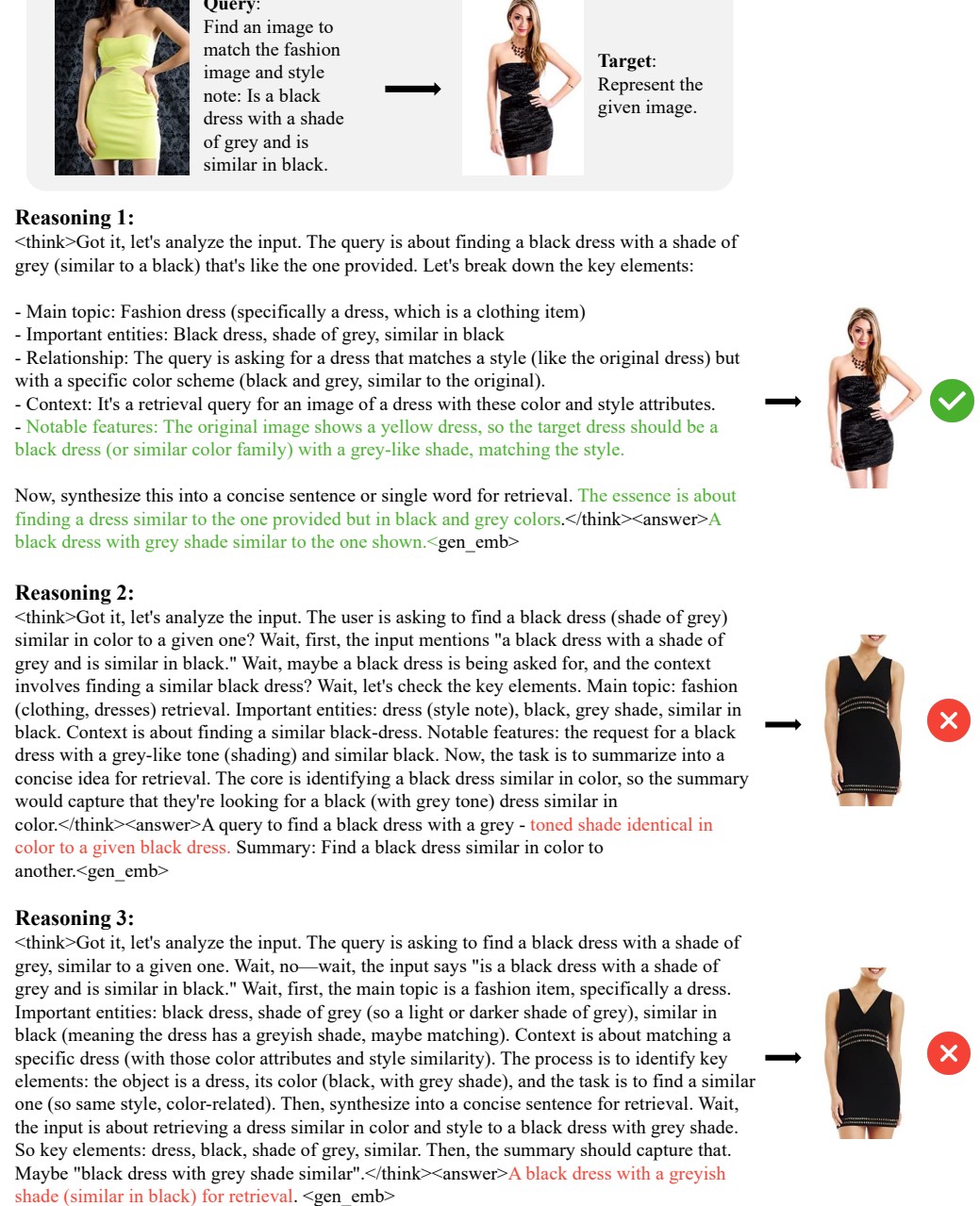

**Query**:
Find an image to match the fashion image and style note: Is a black dress with a shade of grey and is similar in black.

**Target**:
Represent the given image.

**Reasoning 1:**
\<think\>Got it, let's analyze the input. The query is about finding a black dress with a shade of grey (similar to a black) that's like the one provided. Let's break down the key elements:

- Main topic: Fashion dress (specifically a dress, which is a clothing item)
- Important entities: Black dress, shade of grey, similar in black
- Relationship: The query is asking for a dress that matches a style (like the original dress) but with a specific color scheme (black and grey, similar to the original).
- Context: It's a retrieval query for an image of a dress with these color and style attributes.
- Notable features: The original image shows a yellow dress, so the target dress should be a black dress (or similar color family) with a grey-like shade, matching the style.

Now, synthesize this into a concise sentence or single word for retrieval. The essence is about finding a dress similar to the one provided but in black and grey colors.\</think\>\<answer\>A black dress with grey shade similar to the one shown.\<gen_emb\>

**Reasoning 2:**
\<think\>Got it, let's analyze the input. The user is asking to find a black dress (shade of grey) similar in color to a given one? Wait, first, the input mentions "a black dress with a shade of grey and is similar in black." Wait, maybe a black dress is being asked for, and the context involves finding a similar black dress? Wait, let's check the key elements. Main topic: fashion (clothing, dresses) retrieval. Important entities: dress (style note), black, grey shade, similar in black. Context is about finding a similar black-dress. Notable features: the request for a black dress with a grey-like tone (shading) and similar black. Now, the task is to summarize into a concise idea for retrieval. The core is identifying a black dress similar in color, so the summary would capture that they're looking for a black (with grey tone) dress similar in color.\</think\>\<answer\>A query to find a black dress with a grey - toned shade identical in color to a given black dress. Summary: Find a black dress similar in color to another.\<gen_emb\>

**Reasoning 3:**
\<think\>Got it, let's analyze the input. The query is asking to find a black dress with a shade of grey, similar to a given one. Wait, no—wait, the input says "is a black dress with a shade of grey and is similar in black." Wait, first, the main topic is a fashion item, specifically a dress. Important entities: black dress, shade of grey (so a light or darker shade of grey), similar in black (meaning the dress has a greyish shade, maybe matching). Context is about matching a specific dress (with those color attributes and style similarity). The process is to identify key elements: the object is a dress, its color (black, with grey shade), and the task is to find a similar one (so same style, color-related). Then, synthesize into a concise sentence for retrieval. Wait, the input is about retrieving a dress similar in color and style to a black dress with grey shade. So key elements: dress, black, shade of grey, similar. Then, the summary should capture that. Maybe "black dress with grey shade similar".\</think\>\<answer\>A black dress with a greyish shade (similar in black) for retrieval. \<gen_emb\>

Figure 13: An example showing how repeated sampling leads to variations in model-generated reasoning and summaries, resulting in different retrieval outcomes (Case 2). In this example, correctly retrieved cases are able to reason and summarize that the item needs to match the original style, whereas in incorrectly retrieved cases, the reasoning and summarization only consider the color.

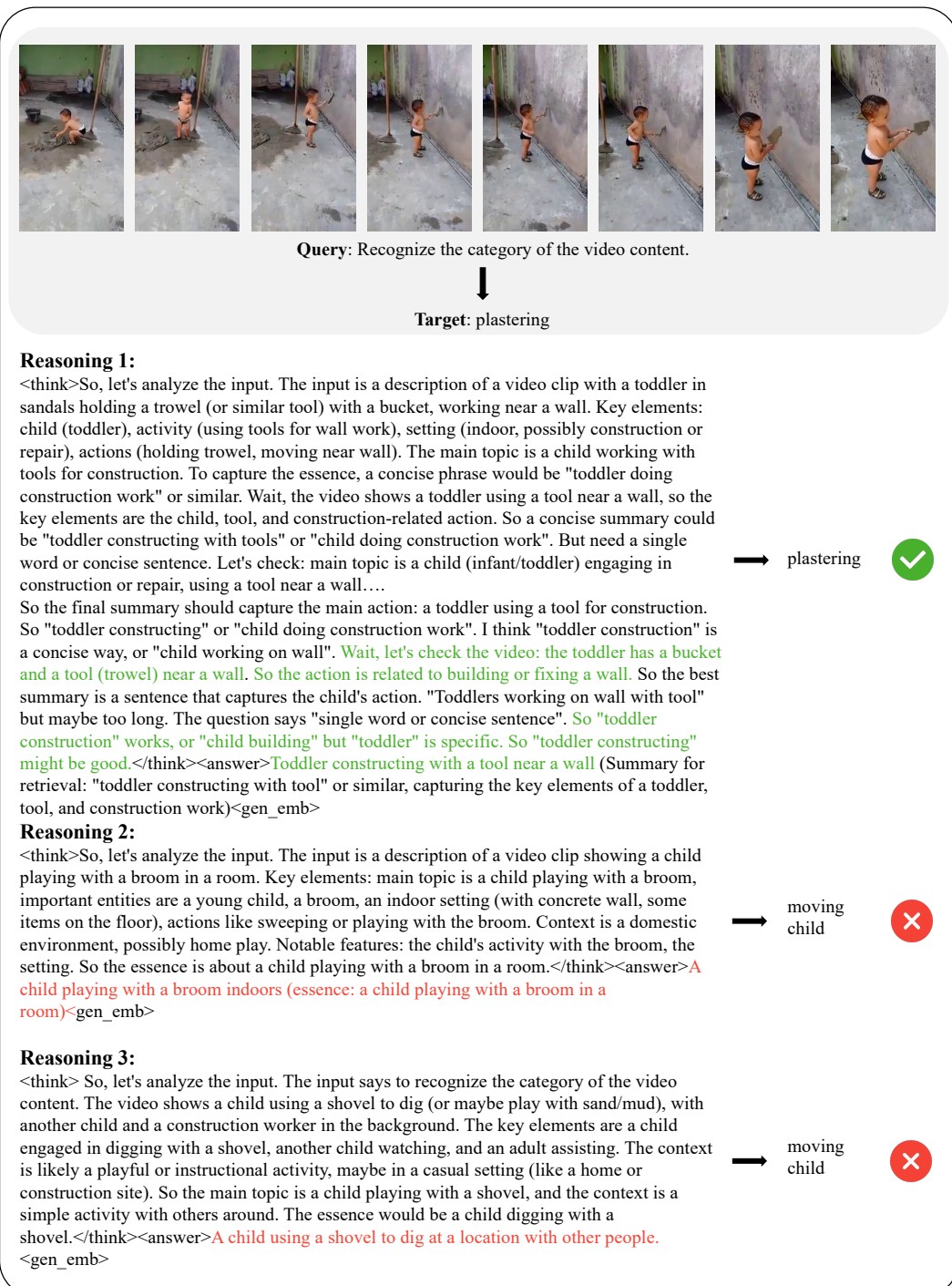

Figure 14: An example showing how repeated sampling leads to variations in model-generated reasoning and summaries, resulting in different retrieval outcomes (Case 3). In this example, only Reasoning Path 1 correctly identifies that the video depicts a child building, leading to the correct retrieval, while the other reasoning paths mainly focus on "playing."

## H  REWARD AND COMPLETION LENGTH VISUALIZATION

In this section, we present visualizations in Figures 15 and 16 illustrating the evolution of reward and completion length throughout training. We observe that for both the 2B and 7B models, the lowest reward value increases as training progresses. However, unlike other tasks, our reward does not exhibit a strictly increasing trend. This is because our RL dataset consists of data from multiple modalities and sources, and follows the VLM2Vec-V2 strategy of using data from the same source within each batch to avoid overly trivial negatives. Due to substantial differences in similarity and difficulty across datasets, the rewards vary considerably between batches: rewards are relatively high when the batch is easier, but lower when the batch is more challenging. Consequently, the reward curve does not follow a strictly monotonic upward trajectory. In addition, we observe that the completion length of the 2B model decreases as training progresses. This trend is consistent with the findings of Chen et al. (2025c), Chen et al. (2025b), and Peng et al. (2025) on small-scale MLLMs. A possible explanation is that the reasoning capacity of the 2B model is limited, and excessively long reasoning may even impair its performance.

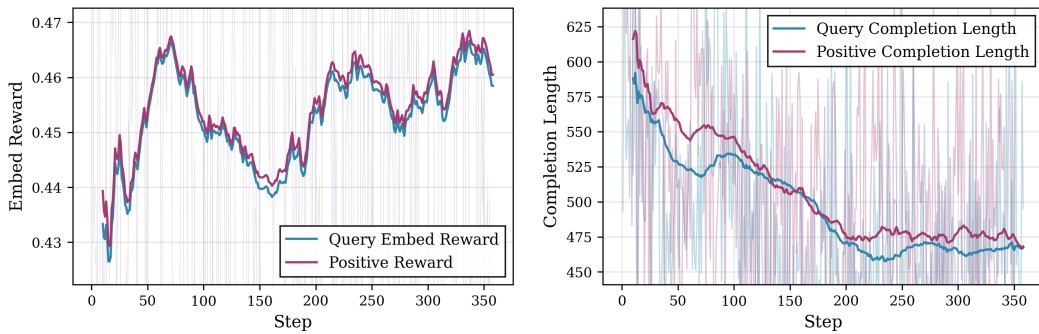

Figure 15: Evolution of reward and generated completion length of UME-R1-2B during training.

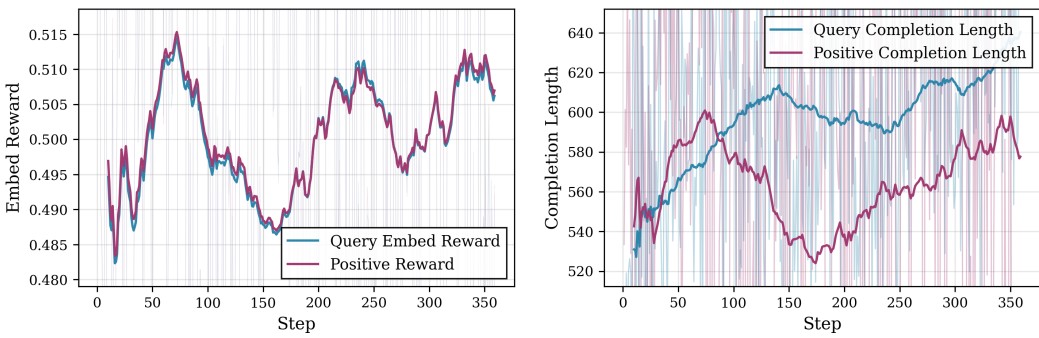

Figure 16: Evolution of reward and generated completion length of UME-R1-7B during training.

