# OpenReview forum: "UME-R1: Exploring Reasoning-Driven Generative Multimodal Embeddings"
_ICLR.cc/2026/Conference — ICLR 2026 Poster_

### Official Review · Reviewer_1Cy2 · 2025-10-29

**Soundness:** 2
**Presentation:** 3
**Contribution:** 2
**Rating:** 4
**Confidence:** 2

**Summary:**

This paper introduces UME-R1, a universal multimodal embedding framework that bridges discriminative and generative paradigms, allowing embeddings to be produced either directly or through reasoning-driven token generation. The method involves a two-stage training process: a supervised stage with chain-of-thought reasoning and summary annotations, followed by reinforcement learning leveraging a novel reward based on template adherence and embedding quality. Evaluations on the MMEB-V2 benchmark demonstrate strong improvements over discriminative embedding models.

**Strengths:**

1. Thorough testing across 78 diverse tasks spanning three visual modalities provides strong empirical validation.
2. Generative embeddings provide interpretable reasoning paths, potentially allowing quality assessment through generated explanations.
3. This paper claims some insights about the relationship between generative and discriminative embeddings as well as their impact on the model's performance.

**Weaknesses:**

1. Generating reasoning and summaries significantly increases inference time and computational requirements compared to traditional discriminative embeddings.
2. The selection mechanism between discriminative and generative embeddings for different tasks is not clearly defined, and there is a gap from the oracle baseline.
3. UME-R1's performance doesn't seem to be comparable with the SOTA baseline on the VisDoc workload.

**Questions:**

1. How is the generated embedding used in different MMEB-V2 tasks?
2. How would one deploy a system that decides between discriminative and generative embeddings in practical applications? Is there any mechanism—heuristic or learned—that enables adaptive switching?
3. UME-R1 doesn't beat the previous SOTA work on the VisDoc workload. Can you provide some potential solutions to improve performance for this specific task?
4. In the ablation study, the author claims that the RL stage improves the performance substantially. However, the improvement seems marginal according to Table 2.

---

> ### Author Response · Authors · 2025-11-21
> **Response to Reviewer 1Cy2 (1/2)**
>
> Thank you for your thoughtful feedback and recognition of our insights. Below, we address the concerns raised in your review.
>
> >**Weakness 1:** Generating reasoning and summaries significantly increases inference time and computational requirements compared to traditional discriminative embeddings.
>
> **Response:** We acknowledge that generating reasoning and summaries increases inference cost compared to traditional discriminative embeddings. However, this is an inherent feature rather than a limitation of the generative paradigm. Generative embeddings enable inference-time scaling—that is, users can flexibly trade computational cost for improved performance by allocating more inference budget (e.g., through longer reasoning or multiple samples). This controllable scaling property is a key advantage of generative embeddings over conventional discriminative approaches. In addition, the improved interpretability and stronger performance make it worthwhile. Note that UME-R1 supports both discriminative and generative modes, allowing users to trade off efficiency and performance according to their needs.
>
> >**Weakness 2 & Question 1:** The selection mechanism between discriminative and generative embeddings for different tasks is not clearly defined, and there is a gap from the oracle baseline.
>
> **Response:** We would like to clarify this point. As described in the evaluation setup of Section 4.1, “*Unless otherwise specified, we use generative embeddings for evaluation*.” Accordingly, all results on MMEB-V2 are obtained using generative embeddings for all tasks.
>
> The oracle setting is included only to illustrate the potential upper bound when both embedding types are available. These results highlight that the two embedding types complement each other, and neither alone achieves the maximum potential.
>
> As noted by Reviewer DYSt, “*To my knowledge, this is the first work to systematically incorporate chain-of-thought reasoning into multimodal embedding generation*,” and Reviewer Lg7o suggested “*provide a future direction of more interpretable, reasoning-driven generative multimodal embeddings.*” Our focus in this work is to demonstrate the feasibility and effectiveness of this new direction. While a practical selector is not implemented in this work, the oracle analysis provides motivation and guidance for future development of such a mechanism.
>
> Besides, the high oracle upper bound also indicates that, in practical usage, providing a switchable option between the two embedding modes could allow users to obtain more satisfactory results depending on the input or preference.
>
> >**Weakness 3:** UME-R1's performance doesn't seem to be comparable with the SOTA baseline on the VisDoc workload.
>
> **Response:** We appreciate the reviewer’s observation. The SOTA model on the VisDoc workload (GME) was trained on approximately **8 million** samples, including a large amount of additional open-source and proprietary visual document data curated by the authors. In contrast, UME-R1 was trained on only **1.46 million** samples (The visual document contains only **180k** data) without access to such extended resources, which naturally leads to a performance gap on VisDoc. Nevertheless, UME-R1 achieves a higher overall average score across all 78 MMEB-V2 tasks, demonstrating its generalization ability across modalities. It is also worth emphasizing that our goal is not to surpass all task-specific SOTA models, but to explore and validate the generative embedding paradigm, which complements existing discriminative approaches and opens a new research direction for multimodal embeddings.

---

> ### Author Response · Authors · 2025-11-21
> **Response to Reviewer 1Cy2 (2/2)**
>
> >**Question 2:** How would one deploy a system that decides between discriminative and generative embeddings in practical applications? Is there any mechanism—heuristic or learned—that enables adaptive switching?
>
> **Response:** Thanks for the thoughtful question. In practical deployment scenarios (e.g., retrieval systems), switching between discriminative and generative embeddings can be handled in several ways:
>
> First, manual switching based on user requirements is straightforward. For instance, if users prioritize efficiency, the system can use discriminative embeddings; if they prefer accuracy or reasoning-aware retrieval, generative embeddings can be used instead.
>
> Second, since our model can produce both discriminative and generative embeddings simultaneously, it is possible to return results from both modes in parallel, allowing users to select the most relevant ones based on their preferences.
>
> Finally, as described in our conclusion section, we agree that adaptive switching is a promising direction. Several possible directions include:
>  a) developing lightweight predictors that leverage the query to dynamically choose between discriminative and generative embeddings;
>  b) constructing training data that enables an MLLM to directly predict whether to use the discriminative or generative embeddings; and
>  c) training a reranker to reorder the merged retrieval results from both discriminative and generative embeddings.
>
> >**Question 3:** UME-R1 doesn't beat the previous SOTA work on the VisDoc workload. Can you provide some potential solutions to improve performance for this specific task?
>
> **Response:** Thank you for the question. We believe there are several concrete directions that can further improve performance on this task:
>
> 1. **Scaling visual-document data**. As discussed earlier, UME-R1 is trained with significantly fewer supervised examples than the SOTA model (only 18% of its total data) and without any closed-source data. In particular, the amount of visual-document training data is only 180k. Therefore, one of the most effective ways to improve VisDoc performance is to synthesize or incorporate additional task-specific visual-document data.
> 2. **Incorporating hard negative mining**. The SOTA model explicitly constructs and injects hard negatives during training to boost performance. UME-R1 does not currently employ this strategy. Introducing hard negatives in both the SFT and RL stages is a promising direction and can further strengthen the model’s capability on VisDoc.
> 3. **Strengthening visual granularity and layout understanding**. VisDoc tasks require fine-grained reasoning over document structure (e.g., layout, tables, small text). UME-R1 currently uses the same vision features across all tasks without VisDoc-specific adaptation. A potential improvement is to incorporate layout-aware visual features, for example, region-level pooling or layout-guided crops to better capture document structure.
> 4. **Task-specific CoT refinement for reasoning-based embeddings**. Since UME-R1 leverages reasoning-driven generative embeddings, another promising direction is to introduce VisDoc-specific CoT cues. These lightweight prompts can guide the model to attend more to layout, OCR cues, or cross-region relationships, improving performance without increasing the overall amount of training data.
>
> >**Question 4:** In the ablation study, the author claims that the RL stage improves the performance substantially. However, the improvement seems marginal according to Table 2.
>
> **Response:** Although the improvement from the reinforcement learning stage is not as large as that from SFT training, we would like to clarify why this gain is still meaningful in our setting.
>
> Unlike typical RL methods applied to single-task or single-domain scenarios (e.g., mathematics), our RL stage operates over **dozens of heterogeneous tasks and modalities**, making it substantially more challenging to design a unified reward function that works well across all tasks. Achieving consistent improvements across such a broad task spectrum is inherently more difficult than optimizing for a single task.
>
> Moreover, the RL stage uses only **11k training examples—about 0.8% of the SFT data**, yet still yields an overall performance improvement of approximately one point, which we believe is far from marginal given the scale and diversity of our evaluation. This demonstrates that RL offers a promising and scalable path for further enhancing multimodal embeddings.
>
> Additionally, RL provides flexibility for future improvement: by constructing a small number of carefully selected hard negative examples, the model can benefit even further. Thus, RL not only contributes measurable gains within our current system but also opens up a clear avenue for continued progress

---

### Official Review · Reviewer_Lg7o · 2025-10-30

**Soundness:** 3
**Presentation:** 2
**Contribution:** 3
**Rating:** 6
**Confidence:** 1

**Summary:**

This paper propose UME-R1, a novel universal multimodal embedding framework which unifies the discriminative and generative embeddings. A two-stage training strategy applies a cold-start supervised fine-tuning for the model with reasoning capabilities. Extensive experiment is conducted across various visual tasks and the proposed method outperforms conventional discriminative embedding models and provide a future direction of more interpretable, reasoning-driven generative multimodal embeddings.

**Strengths:**

1. Generative embeddings are worth studying and have certain commonalities for multimodal tasks. This work explores a reasoning-driven method that is brave and novel.

2. The experiment result is solid with a newly constructed cold-start supervised fine-tuning dataset for embedding training with intermediate reasoning and summaries.

3. Convincing visualization examples are provided to prove the effectiveness of the proposed method.

**Weaknesses:**

1. Lack of a comparison framework between the traditional multimodal embeddings and the proposed method of generative embeddings. Providing this may help better presentation.

2. Visual examples require relatively detailed explanations; otherwise, they may feel complex and difficult to determine the more efficient aspects of the proposed method when viewed directly.

**Questions:**

1. See weakness.

2. The training cost and inference speed between the proposed method and baselines may require further investigation.

Overall, I think this work is solid. Although it is an essential module in MLLM work, multimodal embeddings is a relatively unfamiliar research field to me compared to MLLM, which calls for a better presentation in several respects on this work. So I tend to adjust my score and confidence based on the author's response and other reviewers' opinions.

---

> ### Author Response · Authors · 2025-11-21
> **Response to Reviewer Lg7o**
>
> Thank you for your insightful review and for recognizing both the novelty and the solid results of our work.
>
> >**Weakness 1:** Lack of a comparison framework between the traditional multimodal embeddings and the proposed method of generative embeddings. Providing this may help better presentation.
>
> **Response:** We would like to clarify that a comparison with traditional multimodal embedding methods is indeed included in our work. Specifically, Appendix D (renamed as Appendix F in the updated manuscript) provides a detailed evaluation of our proposed generative embeddings against several representative traditional multimodal embedding models on the MMEB-V1 benchmark. The results show that even the strongest traditional embedding model lags significantly behind MLLM-based approaches, with an overall score gap exceeding 20 points. Given this substantial performance difference, we focused the main experimental tables in the body of the paper on comparisons among strong MLLM-based baselines to better highlight the relative advances of our method within the most competitive and relevant setting.
>
> >**Weakness 2:** Visual examples require relatively detailed explanations; otherwise, they may feel complex and difficult to determine the more efficient aspects of the proposed method when viewed directly.
>
> **Response:** We thank the reviewer for this valuable feedback. In response, we have revised the appendix to include more detailed explanations for the visual examples. Specifically, in Appendix G and H, we now provide more detailed descriptions for each example in the captions, highlighting how our generative embedding method leads to improved performance compared to baselines.
>
> >**Question 1:** The training cost and inference speed between the proposed method and baselines may require further investigation.
>
> **Response:** Thank you for your suggestion.  Following your suggestion, we have added a dedicated section in Appendix C to discuss these aspects in detail.
> We report the full training cost for both SFT and RL stages as follows:
>
> | Model  |      SFT      |       RL      |
> |--------|:-------------:|:-------------:|
> | UME-R1 | 2336 H20 GPU-hours| 1344 H20 GPU-hours|
> | DUME   | 1487 H20 GPU-hours|      None     |
>
> We also provide a comparison of inference throughput between generative and discriminative embeddings across the four datasets (CIRR, FashionIQ, K700, and MSVD). We supplement discussion on the computational cost in Appendix C, and the corresponding results are listed below:
>
> | Model                   |      CIRR      |      FashIQ     |       K700      |      MSVD      |
> |-------------------------|:--------------:|:---------------:|:---------------:|:--------------:|
> | UME-R1 (Generative)     | 1.48 samples/s | 1.14  samples/s | 0.50  samples/s | 1.10 samples/s |
> | UME-R1 (Discriminative) | 20.0 samples/s |  19.1 samples/s | 1.59  samples/s | 28.0 samples/s |
>
> As expected, generative embeddings incur extra computation due to the reasoning chain.
>
>  **Why the Additional Cost Is Still Worth It**
>
> Despite the overhead, we agree with the reviewer DYSt  that the generative approach is worth it due to three practical benefits:
>
> 1. **Stronger performance** across all benchmarks.
> 2. **Better interpretability**, as the intermediate reasoning and summaries explicitly reveal why the resulting embedding is strong or weak.
> 3. **A new scaling route for embedding**: instead of increasing model size, we scale inference-time computation (reasoning/thinking steps), which recent LLM research shows to be a more effective and economical path for capability gains.
> Furthermore, our model naturally supports switching between discriminative and generative embeddings, allowing users to select either higher-performing or more efficient embeddings depending on their requirements.

---

### Official Review · Reviewer_b8RW · 2025-11-01

**Soundness:** 3
**Presentation:** 3
**Contribution:** 3
**Rating:** 6
**Confidence:** 3

**Summary:**

The paper introduces UME-R1, a multimodal embedding framework that unifies discriminative and generative embeddings within a single model. Specifically, it incorporates reasoning supervision during training and extends the traditional discriminative embedding pipeline to produce generative embeddings through a reasoning-augmented generation process. Moreover, it employs reinforcement learning to further enhance representation quality. Experiments on MMEB-V2 demonstrate consistent improvements over baselines.

**Strengths:**

- The integration of generative embeddings into the existing embedding learning paradigm represents a conceptually meaningful extension beyond prior MLLM-based approaches.
- The application of reinforcement learning with verifiable rewards is a well-considered adaptation for optimizing embedding quality.
- Experiments show consistent and notable improvements on the MMEB-V2 benchmark across multiple modalities.

**Weaknesses:**

1. The improvement achieved by the RL stage is marginal (around one point) relative to its additional computational cost.
2. While the oracle setting illustrates the upper bound of the framework, the resulting scores are overly idealized and offer limited practical relevance without an implementable selector.

**Questions:**

1. The authors should discuss the weaknesses mentioned above to provide a constructive view of the method’s effectiveness.
2. What is the additional inference-time cost of generating reasoning and summary tokens compared with standard discriminative embedding extraction?
3. Have the authors examined how varying the loss weighting among contrastive and generative objectives during the SFT stage affects final performance?
4. Have the authors considered implementing a learned or heuristic selector to approximate the oracle results in practice?

---

> ### Author Response · Authors · 2025-11-21
> **Response to Reviewer b8RW (1/2)**
>
> Thank you for your thoughtful comments and for recognizing the significance of our work.
>
> >**Weakness 1:** The improvement achieved by the RL stage is marginal (around one point) relative to its additional computational cost.
> **Response:** Although the improvement from the reinforcement learning stage is not as large as that from SFT training, we would like to clarify why this gain is still meaningful in our setting.
>
> Unlike typical RL methods applied to single-task or single-domain scenarios (e.g., mathematics), our RL stage operates over **dozens of heterogeneous tasks and modalities**, making it substantially more challenging to design a unified reward function that works well across all tasks. Achieving consistent improvements across such a broad task spectrum is inherently more difficult than optimizing for a single task.
>
> Moreover, the RL stage uses only **11k training examples—about 0.8% of the SFT data**, yet still yields an overall performance improvement of approximately one point, which we believe is far from marginal given the scale and diversity of our evaluation. This demonstrates that RL offers a promising and scalable path for further enhancing multimodal embeddings.
>
> Additionally, RL provides flexibility for future improvement: by constructing a small number of carefully selected hard negative examples, the model can benefit even further. Thus, RL not only contributes measurable gains within our current system but also opens up a clear avenue for continued progress
>
> >**Weakness 2:** While the oracle setting illustrates the upper bound of the framework, the resulting scores are overly idealized and offer limited practical relevance without an implementable selector.
>
> **Response:** We acknowledge that the oracle setting serves as an upper bound and may not be directly applicable in practice. However, its purpose is to quantify the potential gain achievable by an ideal mode selector and to demonstrate that the two embedding modes are complementary rather than redundant. Importantly, the high oracle upper bound also indicates that, in practical usage, providing a switchable option between the two embedding modes could allow users to obtain more satisfactory results depending on the input or preference. This validates the design motivation of UME-R1 and provides a clear direction for developing an automatic selector in future work.
>
> >**Question 1:** The authors should discuss the weaknesses mentioned above to provide a constructive view of the method’s effectiveness.
>
> **Response:** Thank you for the constructive feedback. Following your suggestion, we have added a “Limitations’’ section in Appendix B to discuss the weaknesses highlighted above, including the remaining areas where RL can still be improved and the current limits in approaching the oracle upper bound.
>
> >**Question 2:** What is the additional inference-time cost of generating reasoning and summary tokens compared with standard discriminative embedding extraction?
>
> **Response:** Thank you for the insightful comment. We provide a comparison of the inference speed between generative and discriminative embeddings across four datasets (CIRR, FashionIQ, K700, and MSVD). We supplement the discussion on the computational cost in Appendix C, and the corresponding results are listed below:
>
> | Model                   |      CIRR      |      FashIQ     |       K700      |      MSVD      |
> |-------------------------|:--------------:|:---------------:|:---------------:|:--------------:|
> | UME-R1 (Generative)     | 1.48 samples/s | 1.14  samples/s | 0.50  samples/s | 1.10 samples/s |
> | UME-R1 (Discriminative) | 20.0 samples/s |  19.1 samples/s | 1.59  samples/s | 28.0 samples/s |
>
> While generative embeddings introduce extra computation due to the reasoning chain, we agree with the reviewer DYSt that this overhead is “worth it.” The approach provides three practical benefits:
>
> 1. **Stronger performance** across all benchmarks.
> 2. **Better interpretability**, as the intermediate reasoning and summaries make explicit why the resulting embedding is strong or weak.
> 3. **A new scaling route for embedding**: instead of increasing model size, we scale inference-time computation (reasoning/thinking steps), which has been shown in recent LLM research to be an effective and more economical way to improve capability.
>
> Furthermore, our model naturally supports switching between discriminative and generative embeddings, allowing users to select either higher-performing or more efficient embeddings depending on their requirements.

---

> ### Author Response · Authors · 2025-11-21
> **Response to Reviewer b8RW (2/2)**
>
> >**Question 3:** Have the authors examined how varying the loss weighting among contrastive and generative objectives during the SFT stage affects final performance?
>
> **Response:** Yes, In early small-scale experiments, we have examined the effect of varying the loss weighting between the contrastive and generative objectives during the SFT stage. Specifically, we sampled 5k examples from each sub-dataset within the Image modality for training and evaluated on the Image tasks. The contrastive:generative weight ratio varied from 0.1 to 10. The results are shown below:
>
> | Contrastive：Generative | Image-CLS | Image-QA | Image-RET | Image-GD | Overall |
> |:-----------------------:|:---------:|:--------:|:---------:|:--------:|:-------:|
> |           0.1           |    63.0   |   59.1   |    64.9   |   76.4   |   64.0  |
> |           0.2           |    63.4   |   58.6   |    65.5   |   75.9   |   64.1  |
> |           0.5           |    62.7   |   58.2   |    65.1   |   76.3   |   63.8  |
> |            1            |    62.9   |   57.7   |    65.2   |   76.1   |   63.7  |
> |            2            |    61.6   |   55.9   |    63.7   |   73.9   |   62.1  |
> |            5            |    60.5   |   53.9   |    61.9   |   74.0   |   60.6  |
> |            10           |    60.4   |   53.6   |    59.3   |   74.0   |   59.7  |
>
> We observed that when the ratio exceeds 1, the model’s generated reasoning and summaries become less reliable, leading to performance degradation. Ratios below 1 provide slight improvements. Therefore, for simplicity and given this robustness, we did not perform additional hyperparameter tuning during the full-scale training.
>
> >**Question 4:** Have the authors considered implementing a learned or heuristic selector to approximate the oracle results in practice?
> **Response:** In this work, our main focus is to explore generative embeddings and to establish the potential upper bound of the framework through oracle analysis.
>
> As noted by Reviewer DYSt, “*To my knowledge, this is the first work to systematically incorporate chain-of-thought reasoning into multimodal embedding generation*,” and Reviewer Lg7o suggested “*provide a future direction of more interpretable, reasoning-driven generative multimodal embeddings*.” Our focus in this work is to demonstrate the feasibility and effectiveness of this new direction.
>
> Implementing a learning-based or heuristic selector, as described in our conclusion section, is indeed a promising next step. Several possible directions include:
>
>  1. Developing lightweight predictors that leverage the query or target to dynamically choose between discriminative and generative embeddings;
>  2. Constructing training data that enables an MLLM to directly predict whether to use the discriminative or generative embeddings; and
>  3. Training a reranker to reorder the merged retrieval results from both discriminative and generative embeddings.

---

### Official Review · Reviewer_DYSt · 2025-11-05

**Soundness:** 3
**Presentation:** 3
**Contribution:** 3
**Rating:** 6
**Confidence:** 4

**Summary:**

This paper introduces UME-R1, a universal multimodal embedding framework that produces embeddings conditioned on chain-of-thought reasoning. The approach consists of a two-stage training strategy: supervised fine-tuning augments query-target pairs with intermediate reasoning and summaries generated by a thinking-capable MLLM, followed by reinforcement learning that optimizes a reward function based on both ranking and similarity gaps to further enhance embedding quality. The model can flexibly produce either discriminative embeddings (extracted directly from input tokens) or generative embeddings (extracted from tokens following generated reasoning and summaries). Evaluated on the MMEB-V2 benchmark spanning 78 tasks across video, image, and visual documents, UME-R1 demonstrates significant improvements over conventional discriminative embedding models, with oracle analysis revealing substantial complementarity between the two embedding types.

**Strengths:**

1. The core contribution is novel and well-executed. To my knowledge, this is the first work to systematically incorporate chain-of-thought reasoning into multimodal embedding generation, demonstrating that reasoning-conditioned embeddings can substantially outperform standard discriminative embeddings. The idea of having models generate intermediate reasoning before producing embeddings is intuitive and well-motivated.

2. The two-stage training framework is well-designed and clearly presented. The supervised fine-tuning stage effectively teaches the model to produce both embedding types while developing reasoning capabilities, and the reinforcement learning stage introduces a clever reward design that addresses the challenge of applying RL to embedding tasks which lack ground-truth answers. The combined ranking and similarity gap reward is particularly elegant, addressing the zero gradient problem that would arise from using fixed similarity thresholds.

3. The experimental results are strong and comprehensive. UME-R1 achieves consistent improvements across multiple modalities (images, videos, visual documents) and task types. The ablation studies are thorough, demonstrating the value of each component including the RL stage, the dual reward design, and the generative training objective. The oracle analysis showing 3.6-4.3 point improvements reveals interesting complementarity between discriminative and generative embeddings.

4. The paper includes valuable analysis beyond standard benchmarking. The inference-time scaling experiments (pass@k analysis) reveal an intriguing property of reasoning-conditioned embeddings, and the comparison with external reasoning models demonstrates that self-generated reasoning is more effective than externally-provided reasoning. These insights extend beyond simply showing performance improvements.

5. The writing is generally clear and the paper is well-structured, making it easy to follow the methodology and experimental design.

**Weaknesses:**

1. The terminology and framing claims in the abstract and introduction are too broad and potentially misleading. The paper repeatedly claims to "pioneer the exploration of generative embeddings" and positions UME-R1 as introducing the first "generative" embeddings. However, the term "generative embedding" is overloaded and could reasonably describe several existing approaches. For instance, LamRA and UniIR extract embeddings from generative models during the generation process, which some would consider "generative embeddings." What UME-R1 actually introduces are embeddings conditioned on chain-of-thought reasoning that the model generates before producing the final embedding. This is a more specific contribution than introducing "generative embeddings" broadly. The authors should revise their claims throughout the abstract, introduction, and conclusion to more precisely describe their contribution as "reasoning-driven" or "CoT-conditioned" or "reasoning-conditioned" embeddings rather than claiming primacy on all "generative embeddings."

2. The distinction between discriminative and generative embeddings as defined in this paper needs clearer justification. The paper defines discriminative embeddings as those extracted from the last input token and generative embeddings as those extracted after generating reasoning and summaries. However, this distinction is somewhat arbitrary - both involve forward passes through the model and both produce vector representations. The key difference is really whether intermediate reasoning is generated, not whether the embedding process is "discriminative" vs "generative" in any fundamental sense. The paper would benefit from more precise terminology that doesn't overload these already-loaded terms from the broader machine learning literature.

3. The computational costs and practical implications of the approach deserve more discussion. Generative embeddings require generating potentially lengthy reasoning chains (up to 8,192 tokens according to the experimental setup), which substantially increases inference cost compared to discriminative embeddings. Although, I think the performance improvements and potential interpretability make the approach "worth it", it would be nice to consider these ancillary factors in the work.

**Questions:**

My comments are constructive, containing both my critique as well as approaches to resolve the concern.

---

> ### Author Response · Authors · 2025-11-21
> **Response to Reviewer DYSt**
>
> Thank you for your insightful review and for recognizing both the novelty and the strong results of our work.
>
> >**Weakness 1:** The terminology and framing claims in the abstract and introduction are too broad and potentially misleading. The authors should revise their claims throughout the abstract, introduction, and conclusion to more precisely describe their contribution as "reasoning-driven" or "CoT-conditioned" or "reasoning-conditioned" embeddings rather than claiming primacy on all "generative embeddings."
>
> **Response:** Thank you for pointing this out.  We acknowledge that our original terminology may cause misunderstanding and does not clearly highlight the model’s process of generating intermediate reasoning. Following the reviewer’s suggestion, we have revised the abstract, introduction, and conclusion to clearly describe our contribution as “reasoning-driven” generative embeddings. We agree that this more precise terminology better reflects the technical novelty of our work. We again thank the reviewer for the constructive feedback and have updated the wording throughout the paper accordingly.
>
>
>
> >**Weakness 2:** The distinction between discriminative and generative embeddings as defined in this paper needs clearer justification. The key difference is really whether intermediate reasoning is generated, not whether the embedding process is "discriminative" vs "generative" in any fundamental sense. The paper would benefit from more precise terminology that doesn't overload these already-loaded terms from the broader machine learning literature.
>
> **Response:** Thank you for this helpful suggestion.  Following your advice, we have revised the manuscript to use terminology that more accurately reflects the actual difference. Specifically, we define discriminative embeddings as embeddings obtained by directly encoding the input, without generating any intermediate content. In contrast, reasoning-driven generative embeddings are obtained after the model generates intermediate reasoning and embedding tokens.
>
> >**Weakness 3:** The computational costs and practical implications of the approach deserve more discussion. Although, I think the performance improvements and potential interpretability make the approach "worth it", it would be nice to consider these ancillary factors in the work.
>
> **Response:** Thank you for the insightful comment. We provide a comparison of the inference speed between generative and discriminative embeddings across four datasets (CIRR, FashionIQ, K700, and MSVD). We supplement discussion on the computational cost in Appendix C, and the corresponding results are listed below:
>
> | Model                   |      CIRR      |      FashIQ     |       K700      |      MSVD      |
> |-------------------------|:--------------:|:---------------:|:---------------:|:--------------:|
> | UME-R1 (Generative)     | 1.48 samples/s | 1.14  samples/s | 0.50  samples/s | 1.10 samples/s |
> | UME-R1 (Discriminative) | 20.0 samples/s |  19.1 samples/s | 1.59  samples/s | 28.0 samples/s |
>
> While generative embeddings introduce extra computation due to the reasoning chain, we agree with the reviewer that this overhead is “worth it.” The approach provides three practical benefits:
>
> 1. **Stronger performance** across all benchmarks.
> 2. **Better interpretability**, as the intermediate reasoning and summaries make explicit why the resulting embedding is strong or weak.
> 3. **A new scaling route for embedding**: instead of increasing model size, we scale inference-time computation (reasoning/thinking steps), which has been shown in recent LLM research to be an effective and more economical way to improve capability.
>
> Furthermore, our model naturally supports switching between discriminative and generative embeddings, allowing users to select either higher-performing or more efficient embeddings depending on their requirements.

---

> > ### Comment · Reviewer_DYSt · 2025-11-26
> >
> > Thanks to the authors for taking the time to respond to my comments. The new framing, wording, and results leads me to raise my score.

---

> > > ### Author Response · Authors · 2025-11-26
> > > **Official Comment by Authors**
> > >
> > > Dear Reviewer DYSt,
> > >
> > > Thank you for your thoughtful follow-up and for raising your score. We are glad that our responses addressed your concerns, and we sincerely appreciate your time and positive reconsideration.
> > >
> > > Best regards,
> > >
> > > Authors

---

### Author Response · Authors · 2025-12-01
**Summary of Rebuttal**

Dear Area Chair,

We sincerely thank the reviewers for their thoughtful and constructive feedback. We are encouraged that **reviewers recognize the novelty, soundness, solid results, and profound insights of UME-R1**, and that the paper received **8,6,6,4 before the data-leak incident (27 Nov around 22:00). Reviewer DYSt considered that our rebuttal addressed the concerns and raised score (26 Nov, 20:37).** Due to the early termination of the discussion phase following the data-leak, the remaining three reviewers did not participate in the discussion, but we believe that our responses adequately address their concerns. Below we provide a concise summary of our rebuttal and clarifications to support your final assessment.

---

## Consensus Among Reviewers

- **Novelty & Significance**: Reviewers **DYSt**, **b8RW**, and **Lg7o** explicitly highlight that exploring the reasoning-driven multimodal embedding paradigm is **both novel and meaningful**, providing a new direction for multimodal embeddings. Reviewer **1Cy2** noted that generative embeddings **offer interpretable reasoning paths**.
- **Well-designed Two-Stage Training Framework**: Reviewers **DYSt**, **b8RW** agree that the two-stage training (SFT + RL) is well-considered for optimizing embedding.
- **Solid Empirical Results**: **All reviewers** note UME-R1’s robust gains **across 78 tasks, its effectiveness across image, video, and visual-document modalities**, and the value of the oracle analysis showing complementarity of the two embedding types.
- **Valuable Analysis & Insight**: Reviewers **DYSt** and **1Cy2** agreed that the oracle analysis clearly demonstrates the insight and complementarity between discriminative and generative embeddings. Reviewer **DYSt** further noted that the inference-time scaling experiments (pass@k analysis) reveal an **intriguing property** of reasoning-conditioned embeddings. Reviewer **Lg7o** also remarked that the visualization examples are **convincing** and effectively showcase the strength of the proposed method.

---

## Summary of Revisions and Clarifications

- **Presentation Improvements (DYSt, Lg7o)**: We refined the terminology and phrasing throughout the paper and provided clearer explanations and interpretations for all visual examples.
- **Inference Cost (DYSt, b8RW, Lg7o, 1Cy2)**: We added a discussion in Appendix C on both training and inference overhead to address the reviewers’ questions.
- **Limitations Discussion (b8RW)**: We expanded Appendix B with a detailed discussion of the limitations of the current approach.
- **Loss Weighting (b8RW)**: We provided experimental results on loss weighting, demonstrating that our method does not require careful hyperparameter tuning and therefore does not introduce additional hyperparameters.
- **RL Stage Benefits (Re: b8RW, 1Cy2)**: We clarified why the ~1 point improvement from RL is meaningful in a **78-task** MMEB-V2 setting, especially given that RL uses only **0.8%** of the SFT data.
- **Oracle Results & Selector Discussion (b8RW, 1Cy2)**: We discussed feasibility of heuristic or learned selectors and clarified why oracle results remain practical and valuable despite not being directly attainable.
- **VisDoc SOTA Performance Comparison & Future Directions (1Cy2)**: We clarified why achieving SOTA on VisDoc is challenging due to its **reliance on a large amount of additional task-specific and closed-source data**, and we discussed concrete directions for potential improvements. Despite this, UME-R1 achieves a **higher overall average score** across all 78 MMEB-V2 tasks, demonstrating strong generalization ability.

---

We have highlighted all modifications in the revised paper in blue, and we hope the Area Chair will consider our rebuttal and the corresponding clarifications. We believe these additions adequately address the reviewers’ concerns.

Thank you again for your time and effort!

Best regards,

Authors

---

### Meta-Review · Area_Chair_iiDL · 2025-12-26

**Summary:**

This paper proposes UME-R1, a universal multimodal embedding framework that produces embeddings conditioned on chain-of-thought reasoning. Extensive results on MMEB-V2 show consistent gains, insightful analyses. The approach is novel, technically sound, and impactful.
Based on the feedback from reviewers, the decision was made to recommend it for acceptance. We congratulate the authors on their acceptance!
On the other hand, authors should revise the paper taking into account the reviewers' comments, such as the issues and concerns mentioned in Weaknesses.

**Reviewer Concerns:**

Reviewer concerns are terminology overreach, inference cost, marginal RL gains, and oracle practicality. These issues are largely semantic or practical, and they do not undermine the strong empirical evidence, sound methodology, or the central contribution.

**Reviewer Scores:**

Most reviewers rate soundness, presentation, and contribution as good, with three borderline-accept scores and one borderline-reject. Despite mixed confidence, reviewers agree on novelty, strong experiments, and clear improvements. Overall consensus favors acceptance given consistent gains across modalities and tasks.

---

### Decision · Program_Chairs · 2026-01-26

Accept (Poster)